# Discovering temporally compositional neural manifolds with switching infinite GPFA

**Changmin Yu[1]**,[*] **Maneesh Sahani[2], Máté Lengyel[1, 3]***,

[1]Computational and Biological Learning Lab, Department of Engineering, University of Cambridge
[2]Gatsby Computational Neuroscience Unit, UCL
[3]Center for Cognitive Computation, Department of Cognitive Science, Central European University

## Abstract

Gaussian Process Factor Analysis (GPFA) is a powerful latent variable model for extracting low-dimensional manifolds underlying population neural activities. However, one limitation of standard GPFA models is that the number of latent factors needs to be pre-specified or selected through heuristic-based processes, and that all factors contribute at all times. We propose the infinite GPFA model, a fully Bayesian non-parametric extension of the classical GPFA by incorporating an Indian Buffet Process (IBP) prior over the factor loading process, such that it is possible to infer a potentially infinite set of latent factors, and the identity of those factors that contribute to neural firings in a compositional manner at *each* time point. Learning and inference in the infinite GPFA model is performed through variational expectation-maximisation, and we additionally propose scalable extensions based on sparse variational Gaussian Process methods. We empirically demonstrate that the infinite GPFA model correctly infers dynamically changing activations of latent factors on a synthetic dataset. By fitting the infinite GPFA model to population activities of hippocampal place cells during spatial tasks with alternating random foraging and spatial memory phases, we identify novel non-trivial and behaviourally meaningful dynamics in the neural encoding process.

## 1 Introduction

The dominant view of neural coding is that information is represented in the activities of populations of neurons, which supports robust computation (Dayan and Abbott, 2005). Trajectories in the high-dimensional neural space are often constrained to a low-dimensional "neural manifold" (Churchland et al., 2012; Cunningham and Yu, 2014). Hence, to achieve comprehensive system-level understanding of neural mechanisms, it is essential to be exploratory by firstly uncovering the generic latent factors underlying high-dimensional population activities before committing to a fixed set of behavioural variables. Advanced unsupervised representation learning methods, such as latent variable modelling (LVM), have been developed and applied for such neural data analysis (Churchland et al., 2007; Cunningham and Yu, 2014). However, discovering low-dimensional neural manifolds remains a challenging task, not only due to the high variability in neuronal firing, but also because neural representations likely reflect the internal states of the animal, which can vary substantially even while experimentally controlled or observed variables are held constant.

A key limitation of existing LVM methods is the necessity for pre-specifying latent dimensions. This is usually performed through model-selection approaches (Doya, 2007). In the absence of prior knowledge of encoded behavioural covariates underlying neural responses, selecting the latent manifold dimensions based on cross-validation approaches lacks interpretability, and the selection is often sensitive with respect to the sampling process. Alternative approaches based on regularisation methods, such as automatic relevance determination (ARD; Wipf and Nagarajan, 2007; Jensen et al., 2021; Gokcen et al., 2024), require maximum likelihood (ML) learning based on marginalisation over all training samples. These methods thus select a single set of latent factors that most likely account for *all* observations, but not a potentially different set of latent factors for *each* observation.

---

[*]Please send any enquiry to `changmin.yu98@gmail.com` and `m.lengyel@eng.cam.ac.uk`

Here we propose a novel, probabilistically principled model that enables simultaneous posterior inference over the number of latent factors and the set of activated latent factors pertinent to each observation. Specifically, we develop a fully Bayesian nonparametric extension of the Gaussian Process Factor Analysis (GPFA) model (Yu et al., 2008), a popular latent variable model for extracting latent Gaussian process factors underlying population activities over single trials. The resulting model, infinite GPFA, incorporates stochastic activation of latent factors in the loading process, which is modelled by the Indian Buffet Process (IBP) prior (Ghahramani and Griffiths, 2005). The IBP defines a distribution over binary matrices with a finite number of rows and infinite number of columns, hence enabling inference over the potentially infinite number of features, as well as tracking uncertainty associated with factor activations for each observation. Unlike existing methods that assume a fixed loading process, the infinite GPFA is able to infer the temporally dyanmic switching expression of latent factors in each neuron, and this property has important neuroscience implications. As an example, hippocampal place cells exhibit the "dynamic grouping effect", whereby ensemble activities transiently represent spatial locations within different reference frames defined by proximity to corresponding shock zones (Kelemen and Fenton, 2010). Despite stationarity in external stimuli, neural spiking frequently exhibits non-trivial temporal structure, potentially due to differential expression of latent behavioural variables in the population activities given changes in internal states of the animal (Flavell et al., 2022).

Learning and inference in infinite GPFA is performed with tractable variational expectation-maximisation (EM). We exploit the sparse variational approach of Titsias (2009), which significantly improves scalability, making it possible to apply the model to real-world datasets. Through empirical evaluation on synthetic datasets, we show that the infinite GPFA matches the performance of the standard GPFA on a dataset with a deterministic generative process, but significantly outperforms GPFA when variability is introduced to the factor loading process. We apply our model to the population activity of hippocampal place cells recorded during spatial navigation, and identify non-trivial switching dynamics in the neural encoding process, contingent on the engaged task context.

## 2 BACKGROUND

### 2.1 GAUSSIAN PROCESS FACTOR ANALYSIS

GPFA extends standard factor analysis models, by replacing Gaussian factors with Gaussian Process (GP) factors in order to capture non-trivial temporal dependencies in the latent space (Yu et al., 2008). The generative model of GPFA is defined as following (Figure 1a).

$$
\begin{aligned}
f_d(\cdot) &\sim \mathcal{GP}\left(m^d(\cdot), k^d(\cdot, \cdot)\right), && \text{for } d = 1, \ldots, D, \\
\mathbf{h}(x_n) &= \mathbf{C} \cdot \mathbf{F}(x_n) + \mathbf{d}, \quad \mathbf{y}(x_n) \sim p\left(\mathbf{y}(x_n)|\phi(\mathbf{h}(x_n)), \theta\right), && \text{for } n = 1, \ldots, N,
\end{aligned}
\tag{1}
$$

where $m^d(\cdot)$ and $k^d(\cdot, \cdot)$ are the mean and kernel functions for the $d$-th latent factors, respectively[1], $\mathbf{C} \in \mathbb{R}^{M \times D}$ is the loading matrix that projects the latent factors to the neural space, with $M$ being the number of neurons, $\mathbf{d} \in \mathbb{R}^M$ is the offset for the linear transformation, $\mathbf{F}(x_n) = [f_1(x_n) \quad \cdots \quad f_D(x_n)]^T$ is the column-stack of all latent factors at input location $x_n$, $\phi(\cdot)$ is some (non-linear) link function, and $\theta$ represents the set of auxiliary generative parameters.

Beyond the simple case with isotropic Gaussian conditional likelihood and a linear link function, learning and inference in GPFA is generally intractable, especially in neuroscience applications where it is common to assume an exponential link function and conditional Poisson likelihood. Hence, to deal with such intractability, here we describe the generalised procedures for learning and inference based on variational EM. For scalability purposes, we consider extensions of standard GPFA models with sparse-variational approximation based on inducing points (Titsias, 2009; Adam et al., 2016). Inducing points, $\mathbf{U}$, are function evaluations of latent GPs over a selected small number of locations, and can be interpreted as approximate sufficient statistics of the latent processes. Notably, there is a simple linear-Gaussian relationship between $\mathbf{U}$ and $\mathbf{F}$. Letting $\mathbf{x} = [x_1, \quad \ldots, \quad x_N]$ represent a complete set of inputs and $\mathbf{f}_d(\mathbf{x}) = [f(x_1), \quad \ldots, \quad f(x_N)]$ the corresponding values of

---

[1]We assume $m^d(\cdot) = 0$ unless stated otherwise.

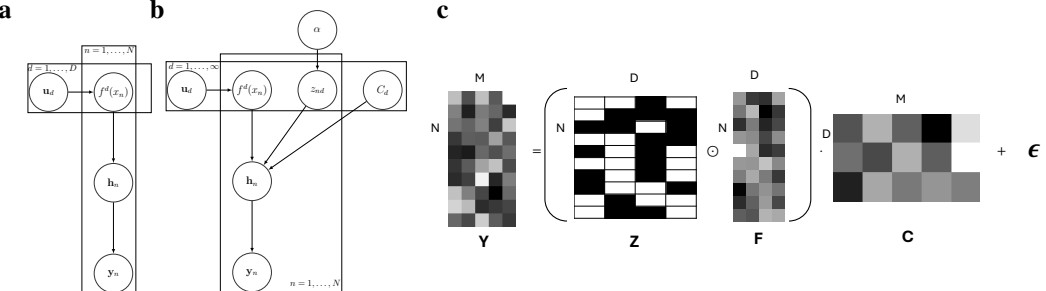

Figure 1: **Graphical demonstration for GPFA and IBP models.** Graphical models for generative processes for standard (**a**) and infinite (**b**) GPFA models with sparse variational approximation. **c**. Illustration of a weighted Gaussian factor analysis model with stochastic binary latent activations. By taking the limit $D \rightarrow \infty$, we essentially place an IBP prior on the binary latent activations, **Z** (Equation 4).

the $d$th latent process, we have

$$p(\mathbf{f}_d(\mathbf{x})|\mathbf{u}_d) = \mathcal{N}\left(\mathbf{K}_{\mathbf{xw}}^d(\mathbf{K}_{\mathbf{wz}}^d)^{-1}\mathbf{u}_d, \mathbf{K}_{\mathbf{xx}}^d - \mathbf{K}_{\mathbf{xw}}^d(\mathbf{K}_{\mathbf{ww}}^d)^{-1}(\mathbf{K}_{\mathbf{xw}}^d)^T\right) \tag{2}$$

where $\mathbf{w}^d$ denotes the $S$ inducing locations for the $d$-th latent process[2], and $\mathbf{K}_{\mathbf{xw}}^d \in \mathbb{R}^{N \times S} = \left[k^d(x_n w_{ds})\right]_{n,s}$.

We choose the joint variational form over **F** and **U** (Titsias, 2009), $q(\mathbf{F}, \mathbf{U}) = \prod_{d=1}^{D} p(\mathbf{f}_d|\mathbf{u}_d)q(\mathbf{u}_d)$, with $q(\mathbf{u}_d) = \mathcal{N}(\boldsymbol{\mu}_d^u, \mathbf{S}_d^u)$. With Equation 2, these assumptions imply a marginal variational approximation for **f** of the form $q(\mathbf{f}_d) = \mathcal{N}(\boldsymbol{\mu}_d^f, \mathbf{S}_d^f)$.

$$\mu_{nd}^f = k^d(x_n, \mathbf{w})\left(\mathbf{K}_{\mathbf{ww}}^d\right)^{-1}\boldsymbol{\mu}_d^f, \quad (s_{nd}^f)^2 = k_{nn}^d + k_{n\mathbf{w}}^d\left((\mathbf{K}_{\mathbf{ww}}^d)^{-1}\mathbf{S}_d^u(\mathbf{K}_{\mathbf{ww}}^d)^{-1} - (\mathbf{K}_{\mathbf{ww}}^d)^{-1}\right)k_{\mathbf{w}n}^d,$$

The corresponding variational free energy objective takes the form,

$$\mathcal{F}[q] = \sum_x \langle \log p(\mathbf{y}|\phi(\mathbf{h}))\rangle_{q(\mathbf{h})} - \sum_{d=1}^{D} \mathrm{KL}\left[q(\mathbf{u}_d)||p(\mathbf{u}_d)\right],$$

Note that $q(\mathbf{h})$ is additively GP-distributed (Equation 1). In general, the expected log conditional likelihood can only be evaluated approximately (Duncker and Sahani, 2018; Keeley et al., 2020). However, it is possible to compute the expected log conditional-likelihood under certain assumptions of conditional likelihood and link function (e.g., Gaussian observation and identity link function, see Section 3). The KL divergence between the variational approximation and GP prior over the inducing points can be evaluated analytically. Both variational and generative parameters can be trained via iterative gradient descent such that the free energy is maximised.

## 2.2 INDIAN BUFFET PROCESS

Standard LVM methods require a pre-set number of latents, which each influence the observations at all times. However, the true number of features is often unknown *a priori*, and the set of active latent factors may vary across different observations. Classical model selection techniques such as cross validation and automatic relevance determination cannot identify observation-specific feature activations. The statistically principled alternative is to perform posterior inference over latent activations given a prior distribution. Here, we consider the Bayesian nonparametric prior known as the Indian Buffet Process (IBP; Ghahramani and Griffiths, 2005), which is a distribution over infinite binary matrices that enables simultaneous posterior inference over the number and identity of latent features underlying each observation.

---

[2]For simplicity, we assume all latent processes share the same set of $S$ inducing locations, **w**.

As a motivating example, we consider a Gaussian factor analysis model with stochastic latent activations (Figure 1c).

$$f_d \sim \mathcal{N}(0, \sigma_d^2), \quad \pi_d \sim \text{Beta}\left(\frac{\alpha}{D}, 1\right), \mathbf{C}_d \sim \mathcal{N}(\mathbf{0}, \sigma^2 \mathbf{I}), \qquad \text{for } d = 1, \ldots, D,$$

$$p(\mathbf{Z}|\boldsymbol{\pi}) = \prod_{d=1}^{D} \pi_d^{m_d} (1 - \pi_d)^{N - m_d}, \quad \mathbf{y}_n = \mathbf{C}(\mathbf{Z}_n \odot \mathbf{F}) + \boldsymbol{\epsilon}_n, \quad \text{for } n = 1, \ldots, N. \tag{3}$$

where $m_d = \sum_{n=1}^{N} \mathbb{1}(z_{nd} = 1)$, and $\odot$ represents the Hadamard product.

Given the conjugacy between Beta and Bernoulli distributions, we can tractably marginalise $\pi$ out.

$$p(\mathbf{Z}) = \prod_{d=1}^{D} \frac{\frac{\alpha}{D} \Gamma(m_d + \frac{\alpha}{D}) \Gamma(N - m_d + 1)}{\Gamma(N + 1 + \frac{\alpha}{D})},$$

Taking the limit $D \to \infty$, the IBP places a prior on $[\mathbf{Z}]$, the canonical form of $\mathbf{Z}$ that is permutation-invariant (Ghahramani and Griffiths, 2005).

$$p([\mathbf{Z}]) = \frac{\alpha^{\mathfrak{D}} \exp(-\alpha H_N)}{\prod_{h \in \{0,1\}^N \setminus \mathbf{0}} \mathfrak{D}_h!} \prod_{d=1}^{\mathfrak{D}} \frac{(N - m_d)!(m_d - 1)!}{N!} \tag{4}$$

where $\mathfrak{D}$ is the number of non-zero columns in $\mathbf{Z}$, $H_N = \sum_{n=1}^{N} \frac{1}{n}$ is the $N$-th harmonic number, $m_d$ is the number of one-entries in the $d$-th column of $\mathbf{Z}$, $\mathfrak{D}_h$ is the number of occurrences of non-zero binary column vector $h$ in $\mathbf{Z}$, $\alpha$ is the prior parameter that controls the expected number of features present in each observation.

A useful alternative interpretation of IBP is based on the stick-breaking construction (Teh et al., 2007), which defines $\pi_d$ as the product of stick-breaking weights, $\pi_d = \prod_{i=1}^{d} v_d$, where $v_d \sim \text{Beta}(\alpha, 1)$. The stick-breaking construction is mathematically equivalent to the Beta-Bernoulli approximation (Equation 3) in the infinite limit, but admits the explicit "sparsity constraint", i.e., the probability of employing the $d$-th latent factor decreases exponentially with $d$, and $\alpha$ controls the expected number of latents.

Inference given the IBP prior can be performed with either MCMC or variational methods (Ghahramani and Griffiths, 2005; Doshi et al., 2009). Here we briefly review the mean-field variational inference approach given the stick-breaking formulation of IBP, outlined in Doshi et al. (2009). Assuming factorisation across different latent factors and observations, the variational distributions for the latent variables (and parameters) are defined as following.

$$q(v_d | a_d, b_d) = \text{Beta}(a_d, b_d), \quad q(\mathbf{C}_d | \boldsymbol{\mu}_d, \mathbf{S}_d) = \mathcal{N}(\boldsymbol{\mu}_d, \mathbf{S}_d), \quad \text{for } d = 1, \ldots, D,$$
$$q(z_{nd} | \tau_{nd}) = \text{Bernoulli}(\tau_{nd}), \qquad\qquad\qquad \text{for } d = 1, \ldots, D, n = 1, \ldots, N,$$

where $D$ is some pre-specified upper bound for the number of latent factors. However, given the sparsity constraint of the IBP prior, the expected number of factors is usually much smaller than $D$, and is controlled by $\alpha$. It is possible to also incorporate a prior over $\alpha$, facilitating more robust and accurate inference of the effective latent dimension (see Section 3). Given the conditional independence within the generative model, the variational free energy objective takes the form[3].

$$\mathcal{F}[q] = \langle \log p(\boldsymbol{\pi}, \mathbf{C}, \mathbf{Z}, \mathbf{Y}) - \log q(\boldsymbol{\pi}) q(\mathbf{C}) q(\mathbf{Z}) \rangle$$
$$= \sum_{d=1}^{D} \langle \log p(\pi_d) \rangle + \sum_{d=1}^{D} \langle \log p(\mathbf{C}_d) \rangle + \sum_{n=1}^{N} \sum_{d=1}^{D} \langle \log p(z_{nd} | \pi_d) \rangle + \sum_{n=1}^{N} \langle \log p(\mathbf{y}_n | \mathbf{Z}_n, \mathbf{C}) \rangle + \mathbb{H}[q]$$

Detailed derivations for the variational objective can be found in Section 3 and Supplemental 1.

---

[3]Unless necessary, we do not explicitly show the variational distributions the expectation is taken with respect to for notational simplicity.

## 3   INFINITE GPFA

Analogous to the motivation behind the original proposal of IBP, we now propose infinite GPFA, the fully Bayesian nonparametric extension of standard GPFA that allows simultaneous inference over the optimal number of latent features and the set of most likely active latent factors underlying *each* observation, whilst capturing non-trivial temporal dependencies within expressions of the set of instantiated latent factors. Specifically, the generative process of infinite GPFA is the following (Figure 1b; details of all derivations in this section can be found in Supplemental S2):

$$
\begin{aligned}
&f_d(\cdot) \sim \mathcal{GP}\left(0, k^d(\cdot, \cdot)\right), && \text{for } d = 1, \dots, D, \\
&v_d \sim \text{Beta}(\alpha, 1), \quad \pi_d = \prod_{i=1}^{d} v_i, \quad \mathbf{C}_d \sim \mathcal{N}(\mathbf{0}, \sigma^2 \mathbf{I}), && \text{for } d = 1, \dots, D, \\
&\alpha \sim \text{Gamma}(s_1, s_2), \\
&z_{nd} | \pi_d \sim \text{Bernoulli}(\pi_d), && \text{for } d = 1, \dots, D, n = 1, \dots, N, \\
&\mathbf{h}(x_n) = \mathbf{C} \cdot (\mathbf{Z} \odot \mathbf{F}(x_n)) + \mathbf{d}, \quad \mathbf{y}(x_n) \sim p(\mathbf{y}(x_n) | \phi(\mathbf{h}(x_n))), && \text{for } n = 1, \dots, N,
\end{aligned}
\tag{5}
$$

Note that here we leverage the stick-breaking construction of the IBP distribution (Ghahramani and Griffiths, 2005). Hence, despite setting an upper bound, the model incorporates a sparsity prior allowing automatic model selection of the most likely number of latents. Moreover, as the scaling parameter, $\alpha$, has a significant effect on the growth of the number of factors, we consider the extended model that integrates over $\alpha$. To this end, we place a Gamma prior over $\alpha$.

Here we perform variational learning using the finite mean-field variational approximations, $q(\mathbf{U}, \boldsymbol{\pi}, \mathbf{Z}) = q(\alpha) \prod_{d=1}^{D} [q(\mathbf{u}_d) q(v_d) q(\mathbf{C}_d) \prod_n q(z_{nd})]$.

$$
\begin{aligned}
&q(\alpha | \xi_1, \xi_2) = \text{Gamma}(\alpha | \xi_1, \xi_2), \\
&q(\mathbf{u}_d | \boldsymbol{\mu}_d^u, \mathbf{S}_d^u) = \mathcal{N}(\mathbf{u}_d | \boldsymbol{\mu}_d^u, \mathbf{S}_d^u), && \text{for } d = 1, \dots, D, \\
&q(v_d | a_d, b_d) = \text{Beta}(v_d | a_d, b_d), \quad q(C_d | \boldsymbol{\mu}_d^C, \mathbf{S}_d^C) = \mathcal{N}(C_d | \boldsymbol{\mu}_d^C, \mathbf{S}_d^C), && \text{for } d = 1, \dots, D, \\
&q(z_{nd} | \tau_{nd}) = \text{Bernoulli}(\tau_{nd}), && \text{for } d = 1, \dots, D, , n = 1, \dots, N,
\end{aligned}
\tag{6}
$$

Note that so far we have assumed deterministic $\mathbf{d}$, but we could in principle integrate over $\mathbf{d}$ with a Gaussian prior (see details in Supplemental S2).

Given the conditional independence in the generative process, the free energy objective takes the following expression.

$$
\begin{aligned}
\mathcal{F}[q] = \sum_{n=1}^{N} \langle \log p(\mathbf{y}_n | \mathbf{F}_n, \mathbf{Z}_n) \rangle &- \text{KL}\left[q(\alpha) || p(\alpha)\right] - \sum_{d=1}^{D} \text{KL}\left[q(\mathbf{u}_d) || p(\mathbf{u}_d)\right] \\
&- \sum_{d=1}^{D} \text{KL}\left[q(v_d) || p(v_d)\right] - \sum_{n,d} \langle \text{KL}\left[q(z_{nd}) || p(z_{nd})\right] \rangle_{q(\mathbf{v})}.
\end{aligned}
\tag{7}
$$

Most terms in the variational free energy admit analytical expressions apart from the expected conditional log-likelihood and the cross-entropy term for the stick-breaking weights, $\mathbf{v}$. Due to the introduction of the binary factor activation matrix, $\mathbf{Z}$, $q(\mathbf{h})$ no longer follows a Gaussian distribution, hence previous approximation approaches based on Gaussian quadrature do not apply (Duncker and Sahani, 2018). Instead, we leverage Taylor expansion for approximating the expected conditional log-likelihood, which offers an effective tradeoff between computational efficiency and approximation accuracy[4]. Specifically, consider exponential link function and Poisson likelihood, we have the following approximation for the expected log-likelihood term.

$$
\langle \log p((\mathbf{y}_n | \mathbf{F}_n, \mathbf{Z}_n)) \rangle \approx \sum_{m=1}^{M} y_{nm} \langle h_{nm} \rangle - \left( \exp\langle h_{nm} \rangle + \frac{1}{2} \text{Var}[h_{nm}] \exp\langle h_{nm} \rangle \right) - \log y_{nm}!
$$

---

[4]It is possible to evaluate the expected log-likelihood analytically under the special case of Gaussian conditional likelihood with identity link function, see details in Supplemental S2

where we have replaced $\langle \exp h_{nm} \rangle$ with its second-order Taylor expansion (see Supplemental S2.2 for details). The variational expectation of **h** can be easily computed due to linearity, and given the mean-field assumption, the corresponding variance can also be analytically computed using the law of total variance (see expression and derivation details in Supplemental S2.2).

The cross entropy term in evaluating the expected KL-divergence with respect to **Z** is as following.

$$\langle \log p(z_{nd}|\mathbf{v}) \rangle = \sum_{i=1}^{d} z_{nd} \left( \sum_{i=1}^{d} \langle \log v_i \rangle \right) + (1 - z_{nd}) \left\langle \log \left( 1 - \prod_{i=1}^{d} v_i \right) \right\rangle, \qquad (8)$$

The second term cannot be evaluated analytically. However, we could replace the expected log-product term with a second-level variational lower bound leveraging Jensen's inequality (see Supplemental S1 and Doshi et al. (2009) for detailed derivations).

The infinite GPFA model is learned via variational EM, iteratively updating the variational parameters (Equation 6), and the generative model parameters (i.e., **C**, **d** and $\alpha$), via gradient-based updates that maximise the free energy objective. In practice, we employ a standard automatic differentiation library for computing the gradients during training (Paszke et al., 2019). It is also possible to incorporate the IBP prior under the finite Beta-Bernoulli approximation (Equation 3), we leave further details in Supplemental S2. We note that due to the fact that we are leveraging variational optimisation instead of MCMC methods for inference, we cannot effectively infer infinite number of latents, but this lead to minimal impact in practice, see further discussion in Supplemental S2.4.

## 4 RELATED WORKS

Discovering low-dimensional neural manifolds underlying high-dimensional population activities is important for understanding neural computations. There exists various extensions of GP-based descriptive models, such as GPFA and Gaussian Process latent variable model (GPLVM), including incorporation of pointwise non-linearity (Duncker and Sahani, 2018; Keeley et al., 2020; Zhao and Park, 2017; Wu et al., 2017), full non-linearity based on neural networks (Ashman et al., 2020; Yu et al., 2022), flexible non-Poisson spike count distributions (Liu and Lengyel, 2021), etc. However, existing methods require model selection techniques for choosing the appropriate latent dimensions, which is largely heuristic and prone to stochasticity arising from cross validation. One notable exception is an extension of GPFA with ARD prior, the Bayesian GPFA (bGPFA; Jensen et al., 2021; Gokcen et al., 2024), which performs automatic pruning of latent dimensions via ML learning. However, an important shortcoming of posterior inference with ARD prior is that the most likely set of latent factors selected is based on marignalisation over all observations, hence overlooking potential variability in latent activations across observations. From the scalability perspective, the bGPFA model leverages the circulant approximation for variational GP approximation over all input locations. Such approximation leads to additional error and elevated computational complexity (Figure 2c, 2e) comparing to the sparse variational approximation used in the infinite GPFA model.

Since its introduction, the IBP prior has been tightly associated with factor analysis models (Ghahramani and Griffiths, 2005), and has been extended to behaviour modeling (Görür et al., 2006), binary matrix factorisation (Meeds et al., 2006), network link prediction (Battiston et al., 2020), etc. The infinite factorial hidden Markov model (HMM) is a prominent extension that incorporates the IBP prior into generative models with explicit temporal dependency structure (Gael et al., 2008). However, the HMM still admits first-order Markovian and linear latent transitions, whereas incorporating GP latents allows simultaneous modeling of all timesteps, in a non-linear and smoothed manner.

The switching linear dynamical system (SLDS) is a popular class of models for parsing behavioural and neural dynamics into simpler alternating states (Linderman et al., 2017). However, the SLDS assumes categorical switching of latent context variables, whereas the infinite GPFA admits a compositional representation of task context, which is simultaneously more computationally efficient and improves interpretability of latent processes (due to the sparsity constraint given the IBP prior). Moreover, in an SLDS, context switching induces changes only in the latent dynamics, hence requiring a non-negligible number of timesteps (depending on the spectral radius of the latent transition operator) for the reflection of context changes in the observation space. In contrast, the compositional nature of factor loading process in the infinite GPFA model allows immediate differential expression of latent processes (that are consistently fitted over all timesteps) into neural activities. Therefore, the infinite GPFA model is more suitable for studying spontaneous neural dynamics in tasks with rapid context-switching structure (Kelemen and Fenton, 2010; Jezek et al., 2011).

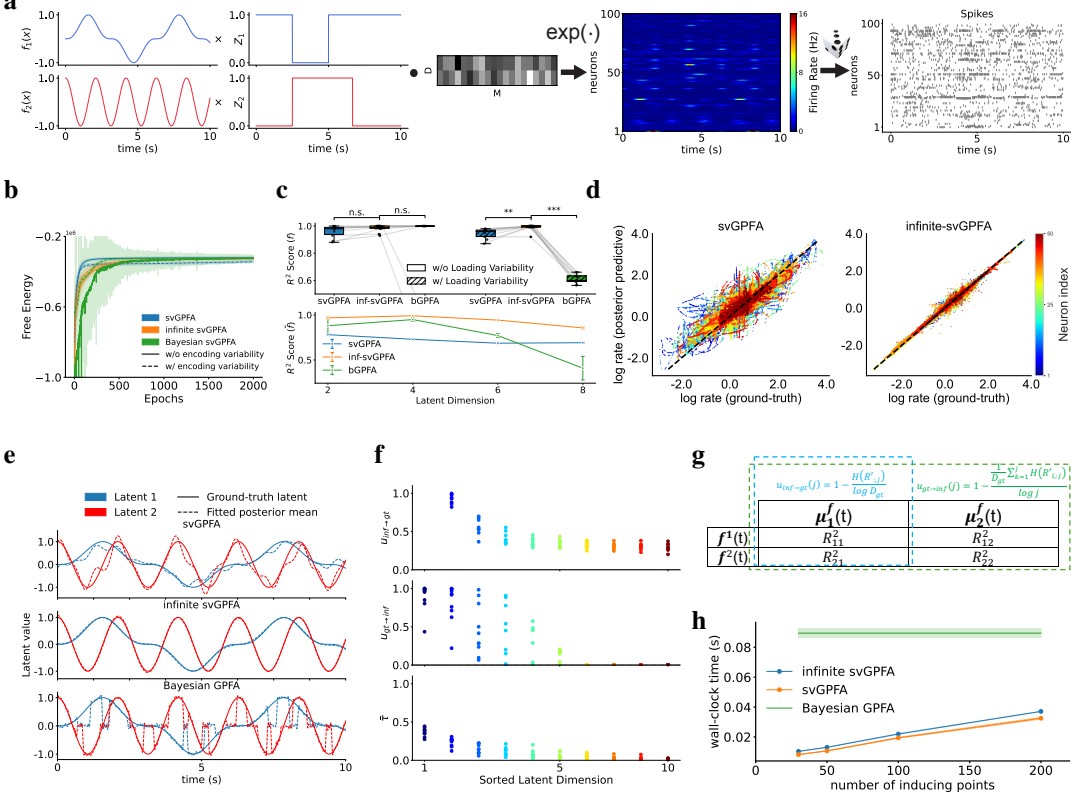

Figure 2: **Empirical valuation of infinite svGPFA on synthetic dataset. a**. Generative process for synthetic data, following the standard GPFA generative model with sinusoidal latent processes ($f_1(x) = \sin^3(x)$ and $f_2(x) = \cos(3x)$) and (optional) binary masking, **Z**. The latents are linearly projected to the neural space, and passed through an exponential link function to generate firing rates, which are then used to generate spikes following time-inhomogeneous Poisson process. **b**. Variational free energy objective during training for different models. **c**. $R^2$ score between posterior means over latent processes and ground-truth latents (top), and loaded latents ($\tilde{\mathbf{f}}$, bottom), for all models, on data given both generative processes (with (bottom) and without encoding variability). **d**. Log-log plot of predicted and ground-truth firing rates for svGPFA (left) and infinite svGPFA (right). Different color represents different neurons. **e**. Comparison of reconstruction of ground-truth latent processes (solid lines) based on best linear-fit given posterior means of latent GPs (dashed lines). **f**. Validating sparseness in inferred latent representations *a posteriori* based on encoding-uniqueness metrics (Equation 9). **g**. Graphical illustration of definition of the uniqueness measures. **h**. Runtime comparison (average wall-clock time of each EM iteration during training). All evaluations are performed based on averaging over 10 random seeds where applicable.

## 5 RESULTS

### 5.1 EMPIRICAL EVALUATION ON SYNTHETIC DATA

We consider a synthetic dataset consists of population spiking generated from two sinusoidal latent processes, following the standard GPFA generative process with exponential link function and Poisson likelihood (Equation 1). To study the effects of variable neural encodings within a single trial, we optionally apply a multiplicative binary mask to the latent processes before projecting them to the neural space (Equation 5). We generate synthetic data for 100 neurons over 10 trials, each lasting 10 seconds in duration, for both cases with and without encoding variability. The data generative process is illustrated in Figure 2a. In the latter scenario, we hand-design the binary mask such that there are continuous periods when one or both processes are actively encoded into the neural space.

Under both generative processes, we fit standard and infinite svGPFA to corresponding population spiking (all hyperparameters are identical for the two models where applicable; values are reported in Supplemental S3). We additionally include the bGPFA model for baseline comparison (Jensen

et al., 2021). All methods converge quickly under either data generative process (Figure 2b). However, it can be clearly observed that the standard svGPFA converges to a suboptimal free energy value when non-trivial binary masking is introduced, whereas the infinite svGPFA reaches a level similar to that achieved with constant latent contributions[5]. Despite reaching similar asymptotic free energy objective, we note the practical concern that the bGPFA training is significantly less stable than the infinite GPFA. Upon training completion, to validate the fidelity of fitted latents, we evaluate and compare the $R^2$ score between the posterior means over the latent processes and the ground-truth latents (Figure 2c, top panel). For the baseline case with trivial binary masks, we observe that all three models perform comparably well, reaching almost perfect discovery of latent processes driving the generation of neural activities. When stochastic factor loading is introduced to the data generative process, we observe that inferred latents of infinite svGPFA predicts ground-truth latents significantly more accurately than those of standard svGPFA (one-sided student-t test, $p = 0.0028$) and bGPFA ($p = 2.17 \times 10^{-16}$). Qualitative inspection of the fitted latents suggests that the bGPFA directly encodes the loaded latents ($\tilde{f}_d(t) = z_t^d f_d(t)$; Figure 2e), which we hypothesise that this is largely due to the circulant approximation of the full covariance matrix over all input locations (instead of linear smoothing given inducing points). We hence evaluate and compare the $R^2$ score between the fitted latents ($\tau_{nd}\boldsymbol{\mu}_{dn}$ for infinite svGPFA, where $\boldsymbol{\mu}_{dn}$ is the mean of the variational distribution over $f_d(\cdot)$ at $x_n$) and $\tilde{\mathbf{f}}$, and observe that the fitting between the bGPFA latents and $\tilde{\mathbf{f}}$ deteriorates quickly as the latent dimension increases, whilst incurring significantly greater computational complexity (Figure 2h). Comparing to the standard svGPFA model, the decline in model fitting is exacerbated in the neural space: infinite svGPFA predicts the log-firing rates (or equivalently, $\mathbf{h}(t)$) significantly more accurately than standard svGPFA when non-trivial masking is present (Figure 2d). The mean squared error of predicted log-rates is $0.40 \pm 0.87$, which is again significantly higher than infinite svGPFA ($0.0043 \pm 0.025$). The absence of explicit mechanisms accounting for such variability in standard GPFA leads to failure in learning the correct generative process due to errors induced by periods when at least one of the factors is not activated, such that erroneous gradients is applied to model parameters due to unexpectedly high prediction errors.

The infinite GPFA model is expected to effectively leverage the sparsity constraints induced by the IBP prior and correctly infer the ground-truth number and identity of latent variables. In order to validate this, we set the number of latent processes in the model, $D$, to be greater than the number of ground-truth latents, $D_{\mathrm{gt}}$, and probe the degree of sparseness within the inferred latent processes. Specifically, we devise the following *encoding-uniqueness metrics* for quantitative evaluation.

$$u_{\mathrm{inf}\to\mathrm{gt}}(d) = 1 - \frac{\mathbb{H}(R'_{:,d})}{\log D_{\mathrm{gt}}} , \quad u_{\mathrm{gt}\to\mathrm{inf}}(d) = 1 - \frac{\frac{1}{D_{\mathrm{gt}}}\sum_{i=1}^{D_{\mathrm{gt}}}\mathbb{H}(R'_{i,:d})}{\log d} , \quad \text{where}$$

$$R'_{:,d} = \left[ \begin{array}{ccc} \frac{R^2(\boldsymbol{\mu}_d^f, \mathbf{f}_1)}{\sum_{i=1}^{D_{\mathrm{gt}}} R^2(\boldsymbol{\mu}_d^f, \mathbf{f}_i)} & \cdots & \frac{R^2(\boldsymbol{\mu}_d^f, \mathbf{f}_{D_{\mathrm{gt}}})}{\sum_{i=1}^{D_{\mathrm{gt}}} R^2(\boldsymbol{\mu}_D^f, \mathbf{f}_i)} \end{array} \right], R'_{i,:d} = \left[ \begin{array}{ccc} \frac{R^2(\boldsymbol{\mu}_1^f, \mathbf{f}_i)}{\sum_{j=1}^{d} R^2(\boldsymbol{\mu}_j^f, \mathbf{f}_i)} & \cdots & \frac{R^2(\boldsymbol{\mu}_d^f, \mathbf{f}_i)}{\sum_{j=1}^{d} R^2(\boldsymbol{\mu}_j^f, \mathbf{f}_i)} \end{array} \right]$$

$$\tag{9}$$

where $\mathbb{H}(\cdot)$ denotes the entropy operator and $R^2(\boldsymbol{\mu}_d^f, \mathbf{f}_i)$ is the $R^2$ score between the posterior mean vector of the $d$-th latent and the $i$-th ground-truth latent. Intuitively, it is expected that latent processes that are most actively loaded into the neural space should encode most information about the underlying processes, and the inferred latents should be encoding only one of the ground-truth latents due to the sparsity constraint induced by the IBP prior. Hence, having sorted the inferred latents based on their averaged posterior responsibilities (Figure 2f, bottom), $u_{\mathrm{inf}\to\mathrm{gt}}(d)$ quantifies how the $d$-th inferred posterior mean uniquely represent one of the ground-truth latent processes (Figure 2g, blue enclosure), and $u_{\mathrm{gt}\to\mathrm{inf}}(d)$ measures how effectively the top $d$ latent processes are representing all ground-truth latents (Figure 2g, green enclosure). Hence, if the model has successfully inferred sparse latent structure with high interpretability, we would expect to observe high uniqueness metric values for latents that are more frequently activated. We indeed observe the expected pattern empirically (Figure 2f, middle and top), indicating incorporating the IBP prior helps with identifying the correct set of interpretable latent processes. To further corroborate our claim, we train infinite svGPFA models with the same number of latent processes as the data generative process, and show that posterior responsibilities of latent processes align well with the ground-truth activation of corresponding latents (Figure S1e).

---

[5]See ablation studies for characterising the effects of latent dimensions and expected sparsity level in Supplemental S3.3.

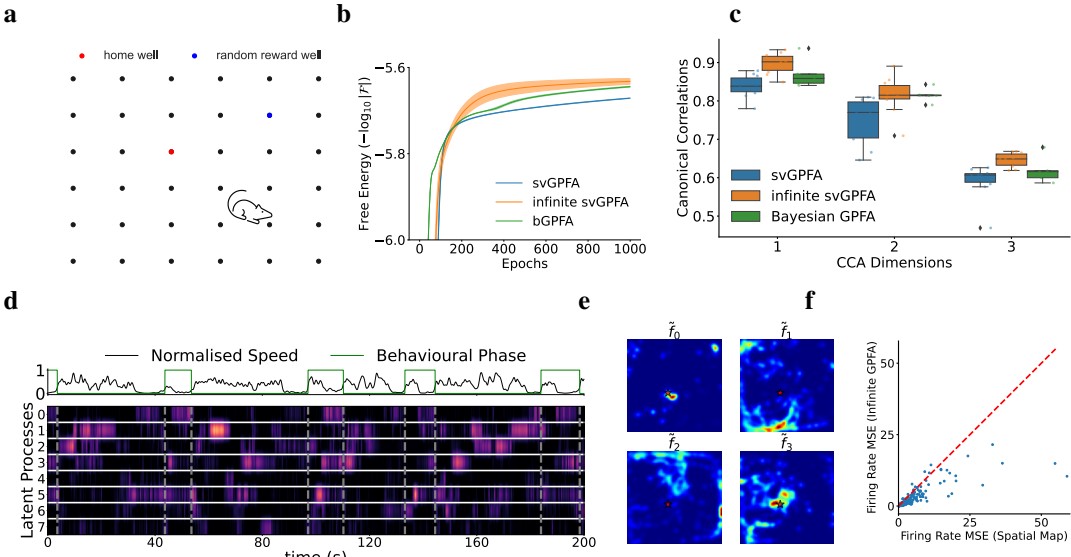

Figure 3: **Probing temporally compositional neural manifolds underlying hippocampal place cell population activities during spatial navigation with alternating behavioural phases. a.** Illustration of behavioural task (Pfeiffer and Foster, 2013). Rats navigate in a $2m \times 2m$ box, with 36 feeding wells uniformly arranged in the box. Animals alternate between searching for reward in a random well (*foraging* phase), and navigate back to a deterministic home well (*homing* phase). **b.** Variational free energy comparison (in log-space) of standard and infinite svGPFA, and bGPFA. **c.** We perform CCA between posterior means over latent processes and selected behavioural variables for the infinite svGPFA model and baseline models. We show comparison of three principal canonical correlations for the two models (dots represent different random seeds). **d.** Temporal trace of posterior responsibilities associated with selected latent processes, with binary behavioural phase (0 and 1 indicate foraging and homing phases, respectively), and normalised speed at the top. **e.** Spatial-tuning of posterior means for different masked latents magnitude ($\tilde{f}_d(x) = \tau_d(x)|\mu_d^f(x)|$). Red star indicates the home well location. **f.** Mean square errors of firing rate prediction given infinite GPFA versus given only spatial location ($p = 3.84 \times 10^{-6}$; one-sample t-test). All evaluations are based on 10 random seeds where applicable.

## 5.2 TEMPORALLY COMPOSITIONAL NEURAL MANIFOLDS UNDERLYING POPULATION FIRING OF HIPPOCAMPAL PLACE CELLS DURING SPATIAL MEMORY TASKS

Having verified that infinite GPFA correctly infers interpretable latent factors and corresponding activations for each observation, we now probe the existence of variability in neural encoding and potential behavioural implications in real neural data. We apply our model to simultaneously recorded population activities of 204 place cells from rat hippocampal CA1, whilst the rat is performing a spatial memory task (Pfeiffer and Foster, 2013). Within each trial, given 36 uniformly arranged feeding wells within a $2m \times 2m$ open-field arena (Figure 3a), rats learn to alternate between foraging for food in an unknown and random location (*foraging* phase), and returning to a fixed home location (*homing* phase). Transitions to the next phase or trial is automatic upon consumption of the reward.

We fit infinite svGPFA to one recording session lasting 2187 seconds, binning spike trains into spike counts within each 30 ms time window. We instantiate maximally 10 latent processes in the model (see empirical comparison of models with varying maximum number of latent factors in Supplemental Figure S2(a)), with 100 inducing points with fixed, equally spaced inducing locations, for each latent factor. Additionally, we fit standard GPFA and bGPFA to the same dataset for baseline comparisons, using the same hyperparameter setting. Through direct inspection of the free energy comparison, we observe the infinite svGPFA yields greater convergence value than standard svGPFA and bGPFA given the same amount of training, suggesting stronger model fitting (Figure 3b).

To validate the fidelity of learned latent factors, we perform canonical correlation analysis (CCA) between posterior means of latent factors and the experimentally verified behavioural correlates of CA1 place cells, including 2-dimensional allocentric spatial location, speed, and head direction of the animal (O'keefe and Nadel, 1978; Geisler et al., 2007; Ormond and O'Keefe, 2022). Similar

to our observation from the evaluations on synthetic dataset (Figure 2c), we found that inferred latents from the infinite svGPFA model provide significantly more faithful explanation for relevant behvaioural covariates than those from standard svGPFA and bGPFA, indicated by the high canonical correlations over the first three principal directions (Figure 3c; additional dimensions yield minimal explained variance of shared covariance given the selected set of behavioural variables, see Supplemental Figure S2(c)). The effective latent dimensions *a posteriori* is approximately 6 (given qualitative inspection of mean posterior responsibility, see Supplemental Figure S2c), coherent with the number of non-trivial principal components within population spiking (Supplemental Figure S2e), and the dimension of latent manifold underlying CA1 place cells population firing reported in previous studies (Nieh et al., 2021; Yu et al., 2022).

We examine activation probabilities for each latent factor at each timestep. Despite stationarity in the marginal distribution of behavioural variables, we observe high temporal variability in posterior responsibilities for all latent processes (Figure 3d), indicating the dynamically compositional nature of low-dimensional neural manifolds underlying population neuronal spikings. By separating the continuous recording into alternating homing and foraging phases, we identify latent processes exhibiting selective activations in accordance with different behavioural phases. Specifically, $f_0(x)$ is activated exclusively during the late periods of both homing and foraging phases, and when the speed is close to $0$ (see top row of Figure 3d). We cross-check with the spatial tuning of corresponding latent factors, computed as the time-average absolute activities of $\tilde{f}_d(x) = \tau_d(x)|\mu_d^f(x)|$ as a function of spatial location of the rat (Figure 3e). Notably, we observe that $\tilde{f}_0(x)$ is almost exclusively selective for the home-well location. The fact that $f_0(x)$ is not only activated upon reaching the home well, but also over the later periods of foraging phase suggests the elevated expression of target location information of the proceeding trial upon reaching the end of the current trial. Such predictive reactivation of goal location information corresponds to goal-oriented hippocampal replay, which is uncovered here in a purely data-driven manner (Diba and Buzsáki, 2007; Pfeiffer and Foster, 2013; Singer et al., 2013). We additionally examine the spatial tuning for other latent processes, and discover that inferred latents exhibits strong spatial selectivity to environmental boundaries and salient landmarks (e.g., home location). Such observation is coherent with the existing theory of spatial coding with hippocampal place cells that leverages combined information of extended cues in given allocentric directions, specifically with respect to the bounding walls (O'Keefe and Burgess, 1996).

Despite the discrepancy in spatial tuning between the latents and place cells, the inferred latents nevertheless represent the spatial location of the animal with high accuracy (Figure S2f). This hence suggests the possibility that the infinite GPFA discovers faithful and generic neural encodings underlying place cells, beyond being merely spatially modulated. We confirm our hypothesis through showing that the firing rate prediction based on infinite GPFA posterior predictive distribution is significantly more accurate than predictions based solely on neuron-specific spatial ratemaps ($p = 3.84 \times 10^{-6}$, one-sample t-test; Figure 3f). Our finding coheres with the well known fact that hippocampal neurons exhibit conjunctive selectivity to multiple behavioural variables (O'keefe and Nadel, 1978; Hardcastle et al., 2017).

## 6 DISCUSSION

We introduce the infinite GPFA, a fully Bayesian nonparametric generalisation of the standard GPFA model. Incorporating the IBP prior over latent activations enables simultaneous inference over both the number of latent factors and the time-varying activation probabilities of factors. Despite having the intrinsic inconsistency that leads to over-estimation of the number of latent factors (Diaconis and Freedman, 1986; Ghahramani and Griffiths, 2005), we provide extensive empirical evaluations on both synthetic and real neural datasets to demonstrate that the infinite GPFA model is capable of performing accurate automatic model selection and learning of sparse and interpretable latent factors. More importantly, we show that utilities of the infinite GPFA model under the context of exploratory investigations of neural data, which we have identified non-trivial coding mechanisms of hippocampal neurons from a purely data-analytical perspective. Here we have suggested a new angle for interpreting neural computations, such that temporal variability within population neuronal activities is attributed to the stochastic loading of latent information, mediated by internal states of the animal (Flavell et al., 2022). Such interpretation could facilitate fundamental understanding of existing experimental data that exhibits changes in neural firings over brief temporal intervals upon context switching, which cannot be explained by existing models that only capture context-dependent dynamics (Kelemen and Fenton, 2010; Jezek et al., 2011; Linderman et al., 2017).

ACKNOWLEDGEMENT

We wish to thank Brad Pfeiffer and David Foster for providing the neural data for analysis in Section 5.2. We also thank Zilong Ji and Neil Burgess, and anonymous reviewers for helpful discussions and feedback on the manuscript. This work was supported by the Gatsby Charitable Foundation (M.S.) and the Wellcome Trust (Investigator Award in Science 212262/Z/18/Z to M.L.).

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

# SUPPLEMENTARY INFORMATION FOR SWITCHING INFINITE GPFA

## S1 MATHEMATICAL DETAILS OF POSTERIOR INFERENCE WITH INDIA BUFFET PROCESS UNDER THE STICK-BREAKING FORMULATION

Here we briefly review posterior inference under the IBP prior with truncated stick-breaking formulation (Doshi et al., 2009). Specifically, we assume the factor analysis generative process, but under the stick-breaking construction 3.

$$
\begin{aligned}
&f_d \sim \mathcal{N}(0, \sigma_d^2), && \text{for } d = 1, \ldots, \infty, \\
&\mathbf{C}_d \sim \mathcal{N}(\mathbf{0}, \nu_d^2 \mathbf{I}), && \text{for } d = 1, \ldots, \infty, \\
&v_d \sim \text{Beta}(\alpha, 1), && \text{for } d = 1, \ldots, D, \\
&\pi_d = \prod_{i=1}^{d} v_i, && \text{for } d = 1, \ldots, D, \quad\quad (\text{S.1}) \\
&p(\mathbf{Z}|\boldsymbol{\pi}) = \prod_{d=1}^{D} \pi_d^{m_d} (1 - \pi_d)^{N - m_d}, && \\
&\mathbf{y}_n = \mathbf{C}(\mathbf{Z}_n \odot \mathbf{F}) + \boldsymbol{\epsilon}_n, && \text{for } n = 1, \ldots, N.
\end{aligned}
$$

We place mean-field variational distributions on the latent variables, $\mathbf{v}, \mathbf{C}, \mathbf{Z}$.

$$
\begin{aligned}
q(v_d | a_d, b_d) &= \text{Beta}(a_d, b_d), && \forall d, \\
q(\mathbf{C}_d | \boldsymbol{\mu}_d, \mathbf{S}_d) &= \mathcal{N}(\boldsymbol{\mu}_d, \mathbf{S}_d), && \forall d, \quad\quad (\text{S.2}) \\
q(z_{nd} | \tau_{nd}) &= \text{Bernoulli}(\tau_{nd}), && \forall n, d,
\end{aligned}
$$

We restate the free energy objective below.

$$
\begin{aligned}
\mathcal{F}[q] &= \langle \log p(\boldsymbol{\pi}, \mathbf{C}, \mathbf{Z}, \mathbf{Y}) - \log q(\boldsymbol{\pi}) q(\mathbf{C}) q(\mathbf{Z}) \rangle \\
&= \sum_{d=1}^{D} \langle \log p(v_d) \rangle + \sum_{d=1}^{D} \langle \log p(\mathbf{C}_d) \rangle + \sum_{n=1}^{N} \sum_{d=1}^{D} \langle \log p(z_{nd} | v_d) \rangle + \sum_{n=1}^{N} \langle \log p(\mathbf{y}_n | \mathbf{Z}_n, \mathbf{C}) \rangle + \mathbb{H}[q]
\end{aligned}
$$

$$(\text{S.3})$$

The key difficulty in evaluating the free energy lies in the computation of the expected log-density of the prior distribution over $\mathbf{Z}$. Conditioning on the stick-breaking weights, the expected log-density of $z_{nd}$ given $\mathbf{v}$ is as following.

$$
\log p(z_{nd} | \mathbf{v}) = \sum_{i=1}^{d} z_{nd} \left( \sum_{i=1}^{d} \log v_i \right) + (1 - z_{nd}) \log \left( 1 - \prod_{i=1}^{d} v_i \right), \quad\quad (\text{S.4})
$$

The first term in the above equation is simple to compute given the Beta variational distribution on $\mathbf{v}$. However, the second term involves the expectation of $\log \left( 1 - \prod_{i=1}^{d} v_i \right)$, which does not admit closed-form expression. To this end, we employ a second-level variational approximation, by leveraging the Jensen's inequality.

$$
\begin{aligned}
\mathbb{E}_{q(\mathbf{v})} \left[ \log \left( 1 - \prod_{i=1}^{d} v_d \right) \right] &= \mathbb{E}_{q(\mathbf{v})} \left[ \log \left( \sum_{l=1}^{d} q_d(l) \frac{(1 - v_l) \prod_{m=1}^{l-1} v_m}{q_d(l)} \right) \right] \\
&\geq \mathbb{E}_{q(\mathbf{v}) q_d(l)} \left[ \log(1 - v_l) + \sum_{m=1}^{l-1} \log(v_m) - \log q_d(l) \right] \quad\quad (\text{S.5}) \\
&= \mathbb{E}_{q_d(l)} \left[ \psi(b_l) + \sum_{m=1}^{l-1} \psi(a_l) - \sum_{m=1}^{l} \psi(a_m + b_m) \right] + \mathbb{H}[q_d(l)],
\end{aligned}
$$

where $q_d(l)$ is a categorical distribution, such that $q_d(l) = q_{dl}$ for all $d$ and $l$. Note that in the above derivations, we leverage the fact that $1 - \prod_{i=1}^{d} v_i = \sum_{l=1}^{d}(1 - v_l) \prod_{m=1}^{l-1} v_m$ on the first line, and Jensen's inequality on the second line.

The lower bound shown in Equation S.5 holds for arbitrary $q_d$, for all $d$. We then take the variational derivative of Equation S.5 with respect to $q_d(l)$, we could derive the optimal $q_d^*(l)$ that maximises the lower bound.

$$q_{dl}^* \propto \exp\left(\psi(b_l) + \sum_{m=1}^{l-1} \psi(a_l) - \sum_{m=1}^{l} \psi(a_m + b_m)\right), \tag{S.6}$$

Note that we need to normalise the above equation such that $q_d^*(l)$ is well-defined.

Substituting Equation S.5 into the computation of the log-conditional prior over $\mathbf{Z}$, we can now optimise the resulting lower bound of the variational free energy objective (Equation S.3).

## S2 VARIANTS OF INFINITE GPFA

### S2.1 VARIATIONAL EXPECTATION-MAXIMISATION FOR INFINITE GPFA WITH GAUSSIAN CONDITIONAL LIKELIHOOD UNDER FINITE BETA-BERNOULLI APPROXIMATION

In the main paper, we exclusively work with the IBP prior under the stick-breaking construction. It is also possible to instantiate the IBP prior under the finite Beta-Bernoulli approximation. With a slight abuse of notations, we term the resulting GPFA model that incorporating the IBP under this construction the "*finite* GPFA" model. The corresponding generative model is shown as following.

$$
\begin{align}
\mathbf{f}_d(\cdot) &\sim \mathcal{GP}\left(0, k^d(\cdot, \cdot)\right), & &\text{for } d = 1, \ldots, D, \tag{S.7a}\\
\alpha &\sim \text{Gamma}(s_1, s_2), & & \tag{S.7b}\\
\pi_d &\sim \text{Beta}(\alpha/D, 1), & &\text{for } d = 1, \ldots, D, \tag{S.7c}\\
z_{nd}|\pi_d &\sim \text{Bernoulli}(\pi_d), & &\text{for } d = 1, \ldots, D, n = 1, \ldots, N, \tag{S.7d}\\
\mathbf{C}_d &\sim \mathcal{N}(\mathbf{0}, \nu_d^2 \mathbf{I}), & &\text{for } d = 1, \ldots, D, \tag{S.7e}\\
\mathbf{h}(x_n) &= \mathbf{C} \cdot (\mathbf{Z}_n \odot \mathbf{F}(x_n)) + \mathbf{d}, & &\text{for } n = 1, \ldots, N, \tag{S.7f}\\
\mathbf{y}(x_n) &\sim p(\mathbf{y}(x_n)|\phi(\mathbf{h}(x_n))), & &\text{for } n = 1, \ldots, N, \tag{S.7g}
\end{align}
$$

Below we show different variants of the generative process presented in Equation S.7.

#### S2.1.1 DETERMINISTIC LOADING MATRIX AND $\alpha$ PARAMETERS

We start with the simplest case by assuming deterministic $\mathbf{C} \in \mathbb{R}^{M \times D}$ and $\alpha \in \mathbb{R}$ (i.e., removing Equations S.7e and S.7b from Equation S.7). Below we show the complete derivation of the

corresponding variational free energy.

$$\mathcal{F}[q]$$

$$= \sum_{n=1}^{N} \langle \log p((\mathbf{y}_n|\mathbf{F}_n, \mathbf{Z}_n)) \rangle - \sum_{d=1}^{D} \text{KL}\left[q(\mathbf{u}_d)||p(\mathbf{u}_d)\right] - \sum_{d=1}^{D} \text{KL}\left[q(\pi_d)||p(\pi_d)\right] - \sum_{n=1}^{N}\sum_{d=1}^{D} \langle \text{KL}\left[q(z_{nd})||p(z_{nd})\right] \rangle_{q(\pi_d)}$$

$$= -\frac{1}{2\sigma^2} \sum_{n=1}^{N}\sum_{m=1}^{M} \left[ \left( y_{nm} - \left( \sum_{d=1}^{D} C_{md}\tau_{nd}\mu_{nd}^{f} + d_m \right) \right)^2 \right.$$

$$\left. + \sum_{d=1}^{D} C_{md}^2 \left( \tau_{nd}^2(s_{nd}^f)^2 + ((\mu_{nd}^f)^2 + (s_{nd}^f)^2)\tau_{nd}(1-\tau_{nd}) \right) \right]$$

$$- \frac{1}{2}\sum_{d=1}^{D} \left( \log \frac{|\mathbf{K}_{\mathbf{ww}}^d|}{|\mathbf{S}_d^u|} - N + \text{tr}\left[(\mathbf{K}_{\mathbf{ww}}^d)^{-1}\mathbf{S}_d^u\right] + (\boldsymbol{\mu}_d^u)^T(\mathbf{K}_{\mathbf{ww}}^d)^{-1}\boldsymbol{\mu}_d^u \right)$$

$$- \sum_{d=1}^{D} \left( \log \frac{\Gamma(a_d + b_d)\Gamma(\frac{\alpha}{D})}{\Gamma(\frac{\alpha}{D}+1)\Gamma(a_d)\Gamma(b_d)} + (a_d - \frac{\alpha}{D})(\psi(a_d) - \psi(a_d + b_d)) + (b_d - 1)(\psi(b_d) - \psi(a_d + b_d)) \right)$$

$$- \sum_{n=1}^{N}\sum_{d=1}^{D} \left[ \tau_{nd}\log\tau_{nd} + (1-\tau_{nd})\log(1-\tau_{nd}) \right]$$

$$+ \sum_{n=1}^{N}\sum_{d=1}^{D} \left[ \tau_{nd}(\psi(a_d) - \psi(a_d + b_d)) + (1 - \tau_{nd})(\psi(b_d) - \psi(a_d + b_d)) \right]$$

$$\tag{S.8}$$

where $\psi(\cdot)$ is the digamma function. For computing the expected conditional log-likelihood, we have leveraged the law of total variance identity.

$$\text{Var}\left[XY\right] = (\mathbb{E}\left[X\right])^2\text{Var}\left[Y\right] + \text{Var}\left[X\right](\mathbb{E}\left[Y\right])^2 + \text{Var}\left[X\right]\text{Var}\left[Y\right], \tag{S.9}$$

### S2.1.2 Incorporating Prior Belief over Loading Matrix $\mathbf{C}$

We then extend the above model by assuming stochastic loading matrix, $\mathbf{C}$, and infer its value given posterior inference. We place Gaussian priors on each row of $\mathbf{C}$, (Equation S.7e), and assume the corresponding variational distributions are also Gaussian.

$$q(\mathbf{C}_d) = \mathcal{N}(\mathbf{C}_d|\boldsymbol{\mu}_d^C, \mathbf{S}_d^C), \text{ for } d = 1, \ldots, D, \tag{S.10}$$

Due to mean-field assumption, this lead to minimal changes to the free energy objective (c.f. Equation S.8). For the quadratic term in the expected log density, we could readily replace $\mathbf{C}_{md}$ with $\mathbb{E}[C_{md}] = (\boldsymbol{\mu}_d^C)_m$. The variance component need a bit more workaround, requiring the law of total variance (Equation S.9).

$$\text{Var}\left[C_{md}Z_{nd}f_{nd}\right] = \mathbb{E}\left[\text{Var}\left[C_{md}Z_{nd}f_{nd}\right]|Z_{nd}, f_{nd}\right] + \text{Var}\left[\mathbb{E}\left[C_{md}Z_{nd}f_{nd}\right]|Z_{nd}, f_{nd}\right]$$

$$= (s_{dm}^C)^2 \left( \tau_{nd}\left( (\mu_{nd}^f)^2 + (s_{nd}^f)^2 \right) \right) + (\mu_{md}^C)^2 \left( \tau_{nd}(s_{nd}^f)^2 + \tau_{nd}(1-\tau_{nd})(\mu_{nd}^f)^2 \right) \tag{S.11}$$

where we have overloaded notations to denote $(s_{dm}^C)^2 = \left(\mathbf{S}_d^C\right)_{mm}$. Moreover, there is an additional KL divergence term between the variational and prior distributions over $\mathbf{C}$, which can be evaluated analytically given the Gaussian parametric assumption.

$$\text{KL}\left[q(\mathbf{C})||p(\mathbf{C})\right] = \sum_{d=1}^{D}\text{KL}\left[q(\mathbf{C}^d)||p(\mathbf{C}^d)\right] = \frac{1}{2}\sum_{d=1}^{D}\left[ M\log\nu_d^2 - \log|\mathbf{S}_d^C| - M + \frac{1}{\nu_d^2}(\text{Tr}(\mathbf{S}_d^C) + (\boldsymbol{\mu}_d^C)^T\boldsymbol{\mu}_d^C) \right] \tag{S.12}$$

Note that so far we have assumed deterministic $\mathbf{d}$, but we can also make it stochastic and place Gaussian priors on it. Comparing with the canonical free energy formulation (Equation S.8), this would lead to the inclusion of an additional KL divergence term between the variational Gaussian approximation and Gaussian prior on $\mathbf{d}$, and replace $d_m$ with $\mathbb{E}[d_m]$, the $m-$ component of mean parameter of the variational distribution over $\mathbf{d}$. The variance component is left unchanged.

### S2.1.3 Incorporating Prior Belief over $\alpha$

The scaling parameter of the IBP prior, $\alpha$, controls the expected number of latent features. Hence, to extend the flexibility of the model, it is preferable to include prior beliefs over $\alpha$ such that it is possible to integrate out $\alpha$, and simultaneously allow posterior inference over $\alpha$ (Escobar and West, 1995; Blei and Jordan, 2004).

Given the non-negative nature of $\alpha$, we place a Gamma distribution over $\alpha$, as well as assume a Gamma variational approximation.

$$q(\alpha|\xi_1,\xi_2) = \text{Gamma}(\alpha|\xi_1,\xi_2) , \tag{S.13}$$

Assuming stochastic $\alpha$ would require replacing terms involving $\alpha$ with corresponding expectations with respect to $q(\alpha|\xi_1,\xi_2)$. In the free energy objective, the only terms involve $\alpha$ is $\text{KL}\left[q(\boldsymbol{\pi})||q(\boldsymbol{\pi})\right]$, we show the updated expression as follows.

$$\text{KL}\left[q(\boldsymbol{\pi})||p(\boldsymbol{\pi})\right] = \sum_{d=1}^{D}\left(\frac{\Gamma(a_d+b_d)}{\Gamma(a_d)\Gamma(b_d)} - \langle\log\alpha\rangle + \log D + \right.$$
$$\left.\left(a_d - \frac{\langle\alpha\rangle}{D}\right)\left((\psi(a_d)-\psi(a_d+b_d))+(b_d-1)(\psi(b_d)-\psi(a_d+b_d))\right)\right) \tag{S.14}$$

Given Equation S.13, we have the following analytical expression for the expectations with respect to $\alpha$.

$$\langle\log\alpha\rangle_{q(\alpha)} = \psi(\xi_1) - \log\xi_2, \quad \langle\alpha\rangle_{q(\alpha)} = \frac{\xi_1}{\xi_2} , \tag{S.15}$$

Additionally, we need to include a KL divergence term between variational and prior distributions over $\alpha$, which can also be analytically computed.

$$\text{KL}[q(\alpha|\xi_1,\xi_2)||p(\alpha|s_1,s_2)] = -\left((\xi_1-s_1)\psi(\xi_1) - \log\Gamma(\xi_1) + \log\Gamma(s_1) + s_1\log\frac{\xi_2}{s_2} + \xi_1\left(\frac{s_2}{\xi_2}-1\right)\right) \tag{S.16}$$

### S2.2 Approximation of Expected Poisson Log-Conditional Density with Exponential Link Function

In the main paper, we predominantly leverage the Poisson conditional likelihood and exponential link function. However, the expected conditional likelihood under such parametric assumption given the IBP-distributed binary mask in the loading process does not admit closed-form expression. We hence leverage second-order Taylor approximation for evaluating the variational expectation of the exponential of log-rate under the generative process.

$$\langle\log p((\mathbf{y}_n|\mathbf{F}_n,\mathbf{Z}_n))\rangle = \sum_{m=1}^{M} y_{nm}\langle h_{nm}\rangle - \langle\exp h_{nm}\rangle - \log y_{nm}!$$
$$\approx \sum_{m=1}^{M} y_{nm}\langle h_{nm}\rangle - \left(\exp\langle h_{nm}\rangle + \frac{1}{2}\text{Var}[h_{nm}]\exp\langle h_{nm}\rangle\right) - \log y_{nm}! \tag{S.17}$$

where we have the expectation and variance of $\mathbf{h}$ with respect to the variational distribution as following.

$$\langle h_{nm}\rangle = \mathbf{C}_m \cdot (\boldsymbol{\tau}_n \odot \boldsymbol{\mu}_n^f) + d_m ,$$
$$\text{Var}[h_{nm}] = \mathbf{C}_m^{\odot 2} \cdot \left(\boldsymbol{\tau}_n^2 \odot (\mathbf{s}_n^f)^2 + ((\boldsymbol{\mu}_n^f)^{\odot 2} + (\mathbf{s}_n^f)^{\odot 2}) \odot \boldsymbol{\tau}_n \odot (1 - \boldsymbol{\tau}_n)\right) \tag{S.18}$$

where $\boldsymbol{\mu}_n^f$ and $(\mathbf{s}_n^f)^2$ are the mean and diagonal-variance of $\mathbf{F}(x_n)$, respectively, and $\odot 2$ represents the elementwise square operation (see Equation S.11).

## S2.3   Approximation of Expected Gaussian Log-Conditional Density

In the main paper, we show how to leverage second-order Taylor expansion for approximating the expected log-conditional density for Poisson likelihood and exponential link function. Such approximation is sufficiently general, and we can increase the order of approximation at the cost of increasing computational complexity. However, here we show a special case for Gaussian likelihood (with deterministic diagonal covariance matrix) and identity link function where we could analytically evaluate such expectation.

$$
\langle \log p((\mathbf{y}_n|\mathbf{F}_n, \mathbf{Z}_n)) \rangle = -\frac{1}{2\sigma^2} \sum_{m=1}^{M} \langle (y_{nm} - h_{nm})^2 \rangle = -\frac{1}{2\sigma^2} \sum_{m=1}^{M} \left( (y_{nm} - \langle h_{nm} \rangle)^2 + \mathrm{Var}[h_{nm}] \right) ,
$$
(S.19)

The expectation and variance of $\mathbf{h}$ with respect to the variational distribution can be computed following Equation S.18.

## S2.4   Further note on the "infinite-ness" in infinite GPFA

Despite what the name suggests, the infinite GPFA model currently does not support instantiation of an effectively infinite number of latent factors. This is due to the fact that we are leveraging variational inference for fitting model, which requires placing a finite upper bound on the number of factors for tractable inference. However, this leads to minimal negative impacts in practice since the intrinsic dimensions of neural manifolds are generally low. Importantly, we wish to emphasise that the main benefit of incorporating the IBP prior is that the Bayesian nonparametric induces the additional flexibility for including new factors *only if* necessary, and enables temporally varying compositional expression of instantiated factors.

# S3   Further Details on Empirical Evaluation

Python implementation of the infinite GPFA model can be access through this repo: `https://github.com/changmin-yu/infinite_gpfa`.

## S3.1   Hyperparameter Settings

All models are trained with Adam optimiser (Kingma and Ba, 2014), with learning rate 0.01. For the main experimental evaluations, we train all models over 2000 epochs. The Bayesian GPFA implementation is based on the official Github repository[6]. All evaluations are based on averaging over 10 random seeds where applicable.

**Synthetic Data.** We instantiate both the standard GPFA and infinite GPFA models with stochastic $\mathbf{C}$, where $\nu_d^2 = 0.1$, the infinite GPFA model further places a Gamma prior on $\alpha$, with $s_1 = 1.0$ and $s_2 = 1.0$ (Equation S.13). We set the number of inducing points to be 30 for the main evaluations, and the corresponding inducing locations are randomly initialised and treated as learnable parameters. For all models, we use the squared exponential (SE) kernels, with trainable scale and lengthscale parameters.

$$
k^d(x, x') = s_d^2 \exp\left(-\frac{||x - x'||}{\tau_d^2}\right) ,
$$
(S.20)

The initial scale and lengthscale parameters are $s_d^0 = 1.0$ and $\tau_d^0 = 0.005$ (in time domain) for all models. For all implemented models, we set the latent dimensions, $D$, to be 10.

**Neural Data.** We preprocess the spiking train data into spike counts, with $30ms$ time window. The instantaneous firing rates for each neuron are computed via dividing the spike counts by the time window size, followed by Gaussian smoothing. The loading matrix, $\mathbf{C}$, is assumed to be deterministic, hence is treated as model parameter and is learned through the variational M-step. The

---

[6]`https://github.com/tachukao/mgplvm-pytorch`

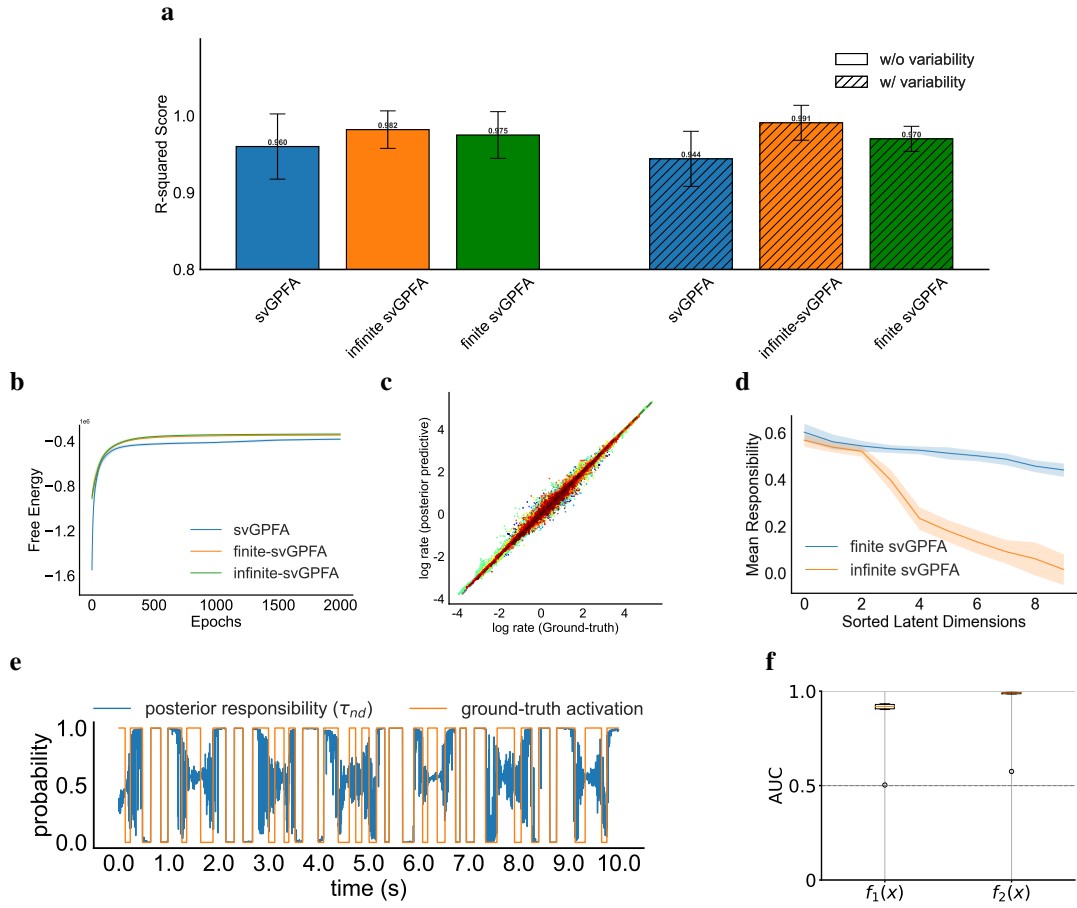

Figure S1: **Further empirical evaluations of infinite svGPFA on the synthetic dataset. a**. Comparison of $R^2$ score between posterior mean of latent processes and ground-truth latents for all three models (standard svGPFA (blue), infinite svGPFA (orange) and finite svGPFA (green)). **b**. Free energy comparison between the three models. **c**. Log-log plot of predicted firing rates against ground-truth firing rates for fitted finite svGPFA model. **d**. Mean responsibility comparison between finite and infinite svGPFA. **e**. Temporal trace of posterior responsibility for a latent process and the ground-truth activation for the corresponding latent (up to permutation). **f**. Area Under Curve (AUC) for binary classification of latent activation given posterior responsibilities.

concentration parameters, $\alpha$, is again assumed to be stochastic, with Gamma prior and parameters $s_1 = 1.0$, $s_2 = 1.0$. For all models, the number of inducing points are 100, and corresponding inducing locations are fixed as equally spaced location along the input (time) domain. We use the SE kernels for the latent GPs with trainable scale and lengthscale parameters. For all implemented models, we set the latent dimensions, $D$, to be 20.

Starting and end times for foraging and homing phases are provided as part of the data collected by the experimenter (Pfeiffer and Foster, 2013).

## S3.2 FURTHER EMPIRICAL EVALUATION ON SYNTHETIC DATA

In addition to the standard and infinite svGPFA with stick-breaking formulation, we also implement and evaluate the *finite* svGPFA model (under the finite Beta-Bernoulli approximation of the IBP prior) on the same synthetic data in the main paper (Figure 2a).

From Figure S1a, we observe that the finite svGPFA model yields stronger performance than the standard svGPFA model, in terms of the $R^2$ score of the linear fitting between the posterior mean of

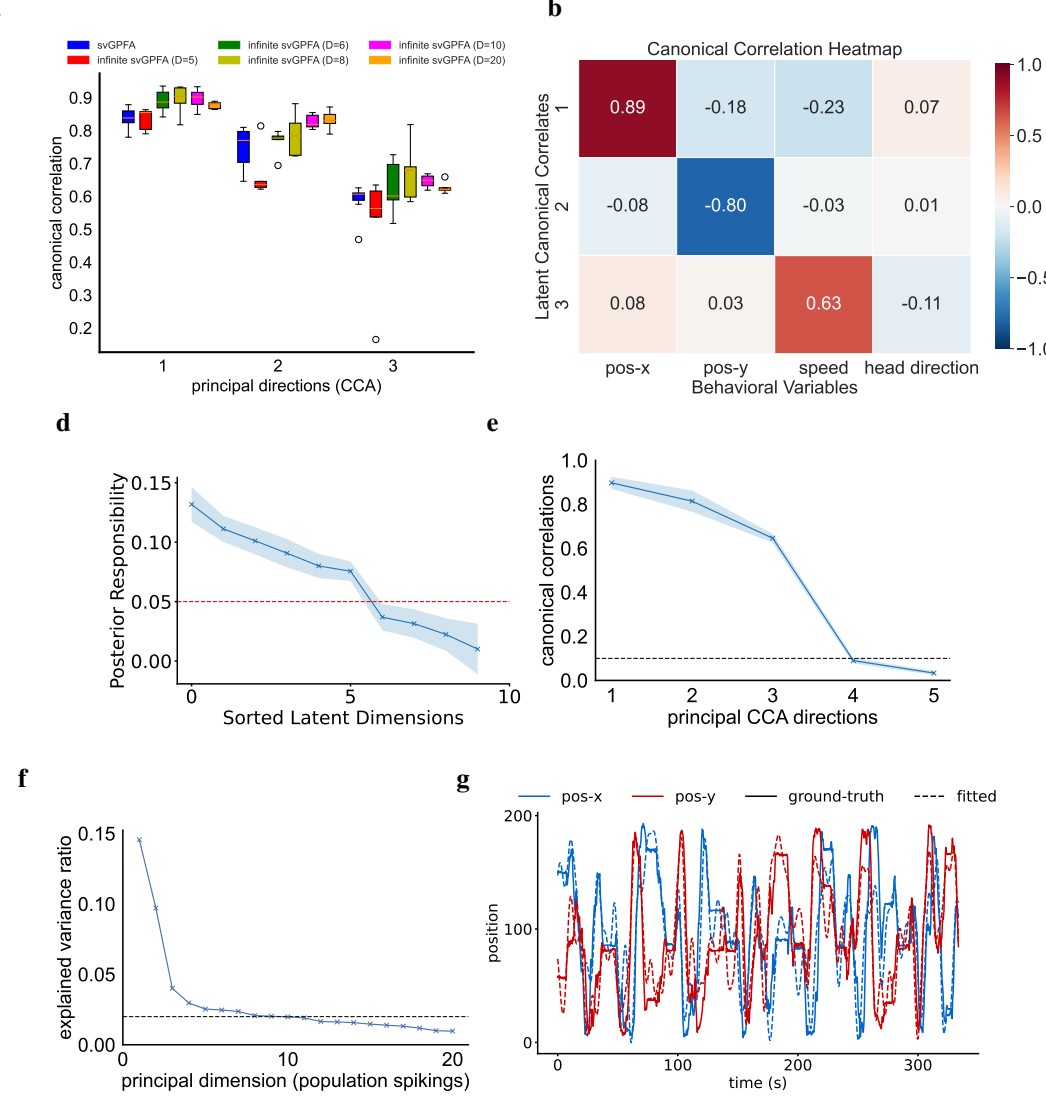

Figure S2: **Further empirical evaluations for fitting infinite svGPFA to the neural data. a**. First three canonical correlations between the posterior mean of the latent processes and the selected behavioural variables, for standard svGPFA, and infinite svGPFA with varying numbers of maximally allowed number of latent factors. **b**. Correlation heatmap between canonical correlates of inferred latents and relevant behavioural variables. **c**. Mean posterior responsibilities (sorted). (d). Canonical correlations between posterior mean of the latents and the set of selected behavioural variables, over the first 5 principal directions. **e**. Explained variance ratios over the top 20 principal directions underlying the population spiking data. **g**. Ground-truth (solid) versus predicted (dashed) x- and y-position on heldout dataset.

the latent processes and the ground-truth latents. We do observe a slight performance drop comparing to the infinite svGPFA, but such drop is insignificant ($p = 0.094$). Similarly, by inspecting the free energy comparison and the log-log plot between the ground-truth firing rates and the predicted firing rates given the finite svGPFA model, we observe that the finite svGPFA leads to similar model fitting performance (Figure S1b, S1c).

The finite svGPFA, due to the finite Beta-Bernoulli formulation, all factor probabilities $\pi_d$ follows the same Beta distribution ($\pi_d \sim \text{Beta}\left(\frac{\alpha}{D}, 1\right)$). Hence, unlike the stick-breaking construction, we do not expect the finite svGPFA to instantiate explicit sparseness constraints, hence does not have the capability to perform automatic model selection for the number of latent factors. Indeed, by inspecting the mean posterior responsibilities associated with the latent factors, we observe that the

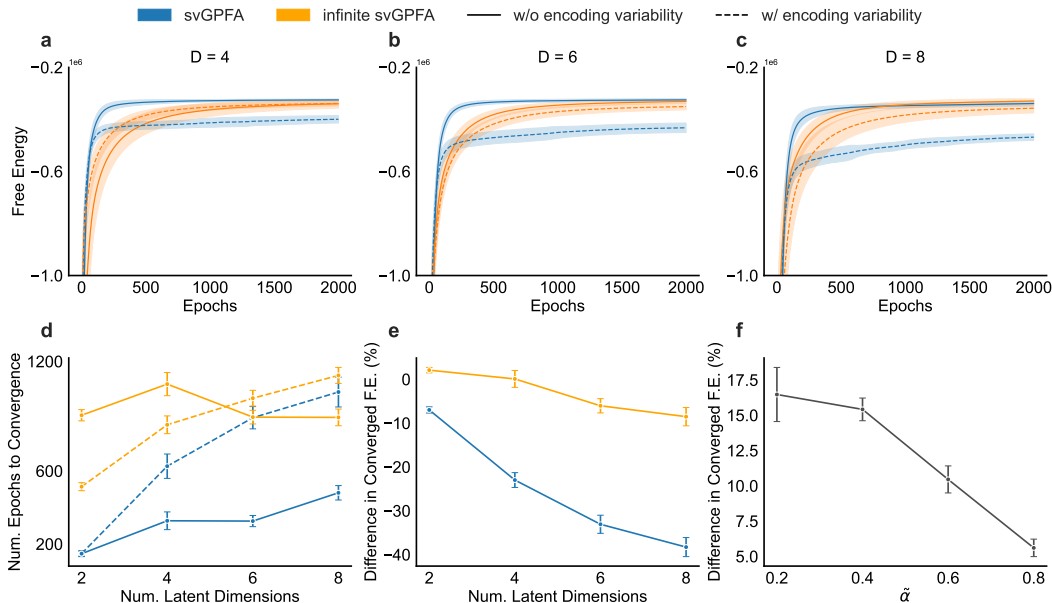

Figure S3: **Ablation Studies on Infinite GPFA on Synthetic Data. a-c**. Temporal trace of free energy during training for standard and infinite svGPFA models, under conditions with and without encoding variability, for varying latent dimensions. **d**. Number of training epochs to reach convergence (**d**) as a function of latent dimensions. **e-f**. Difference (in percentage) between converged free energy objective values between standard and infinite svGPFA, for varying latent dimensions (**e**), and varying expected sparsity level in the loading process (**f**).

finite svGPFA essentially instantiates all latent processes for (differentially) representing ground-truth latents, whereas the infinite svGPFA effectively leverages the top factors instead of attributing the explanatory power to all latent processes (Figure S1d).

To corroborate our analysis in Figure 2, we train infinite svGPFA models with the same number of latent factors as the data generative process. After fitting, we compare temporal traces of posterior responsibilities of inferred latents and corresponding ground-truth activations (up to permutation). We observe that the posterior responsibilities are well-aligned with the actual activations, suggesting the model has correctly identified the dynamic loading of latent factors onto the neural space (Figure S1e, S1f).

### S3.3 ABLATION STUDIES OF INFINITE GPFA ON SYNTHETIC DATA

In the main paper, we have verified that the infinite GPFA outperforms standard GPFA model when there exists non-trivial multiplicative binary masking in the loading process (Figure 2). However, there exists a potential limitation given observations from empirical evaluations: the infinite GPFA models take significantly more training epochs than the standard GPFA models to reach convergence (Figure 2b, 3b). Practical application of the proposed model would become prohibitive if such lagging in convergence scales positively with the latent dimension. We validate this by comparing the temporal trace of the training objective between standard and infinite GPFA models, whilst varying the number of latent processes (Figure S3a-c). Specifically, the (maximally) 8 latent processes are as following.

$$f_1(x) = \sin^3(x), \quad f_2(x) = \cos(3x), \quad f_3(x) = \sin(3x), \quad f_4(x) = \cos^3(x),$$
$$f_5(x) = \sin(x)\cos(2x), \quad f_6(x) = \sin(2x)\cos(x),$$
$$f_7(x) = \sin^2(2x)\cos(x), \quad f_8(x) = \cos^2(2x)\sin(x),$$

(S.21)

We quantify the amount of training to reach convergence by computing the number of training epoch to reach 95% of the asymptotic free energy value. Quantitatively, we indeed observe that the infinite GPFA takes more training epochs to reach convergence (Figure S3d). However, under the trivial-

masking condition, the amount of training for the infinite GPFA model to reach convergence does not increase with the number of latent processes. Additionally, under the condition where there is non-trivial binary masking, despite there exists a positive correlation between the amount of training to reach convergence and the number of latent processes, the magnitude of the positive correlation for the infinite GPFA model is lower than the standard GPFA model. Importantly, as the number of latent processes increases, the gap in the converged free energy objective (under the condition where there is non-trivial binary masking) is significantly greater for the standard GPFA model (Figure S3e)[7]. Collectively, the convergence rate should not be a bottleneck in practical application of the infinite GPFA model, and the significant performance difference as the latent dimension increases yields the infinite GPFA model as the more favourable option empirically.

We additionally verify the effect of expected sparsity in the binary mask. We define the sparsity as the expected number of $1$ (active) entries in the binary mask. Cohering with our expectation, the performance difference between the standard and infinite GPFA ($\frac{\mathcal{F}^{\text{inf-svGPFA}} - \mathcal{F}^{\text{svGPFA}}}{|\mathcal{F}^{\text{svGPFA}}|} \cdot 100$) decreases as the expected sparsity in the mask increases (Figure S3f).

### S3.4 FURTHER EMPIRICAL EVALUATION ON NEURAL DATA

Given trained infinite svGPFA model on the neural data, we choose the effective latent dimension through inspecting the temporal mean of posterior responsibilities associated with the inferred latent processes (Figure S2c). By choosing the threshold of $0.05$, we identify $6$ non-trivially activated latent processes, which is indeed coherent with results reported in literature (Nieh et al., 2021; Yu et al., 2022). To further corroborate the discovery, we perform standard cross validation model selection for selecting the optimal number of latent variables. We use the CCA correlations as the evaluation metrics, we observe that model performance saturates when $D$ reaches $6$ (Figure S2a). Moreover, through directly performing PCA on the population spiking, we observe that the majority of the variance is explained by its first 6-7 principal components (Figure S2e). These observations provide additional empirical validity for the correctness of the optimally inferred number of latents given the IBP prior.

In Section 5.2 of the main text, given the selected set of behavioural variables, we focus our analysis on the first three canonical correlates (Figure 3c). This decision can be justified by the empirical observation that there are only three dominant dimensions explaining the covariance between the latent processes and the selected behavioural variables (Figure S2d).

---

[7]We quantify the gap as the percentage of decrease in free energy at convergence given existence of trivial and non-trivial binary masking, $\frac{\mathcal{F}^{\text{non-trivial}} - \mathcal{F}^{\text{trivial}}}{|\mathcal{F}^{\text{trivial}}|} \cdot 100$

