# OpenReview forum: "Discovering Temporally Compositional Neural Manifolds with Switching Infinite GPFA"
_ICLR.cc/2025/Conference — ICLR 2025 Spotlight_

### Official Review · Reviewer_swfp · 2024-11-03

**Soundness:** 4
**Presentation:** 3
**Contribution:** 3
**Rating:** 8
**Confidence:** 5

**Summary:**

the authors introduce switching infinite gpfa — an extension of the classical gpfa model to account for the possibly time varying dependence of observed neural activity on different latent factors.  the authors make this possible by using an indian buffet process as the generative model for a (infinite) binary mask that selects the latent features read out  to the observation space at each point in time.  they outline how to perform tractable variational EM inference/learning for this model class.  the authors then validate their model and inference/learning procedure on synthetically generated data, and then show how their method can be used to extract behaviorally meaningful latent features from rats performing a spatial navigation task.

**Strengths:**

the paper is very well written. the background section is clear and in my opinion and succesfully takes the reader from the original gpfa to their new generative model that incorporates an indian buffet process prior over the binary masking matrix, Z.   since approximate inference in this model is highly non-trivial, the authors developed an approximate variational EM procedure for inference/learning.  i appreciated the extensive discussion covering what terms are and are not tractable in the variational bound and how the authors deal with the intractable terms in a practical manner;  important details that would clutter the main text were referenced often and helped with further clarifications. their synthetic data example validates their inference approach and reassuringly shows the infinite gpfa model can match standard gpfa inference quality even when there is no encoding variability.  in their last experiment, they apply their method to neurophysiological recordings taken from a rat performing a spatial navigation task;  they demonstrate how their method can reveal the compositional structure of the latent space by identifying different latent factors being loaded onto the neural population at different locations or trial events.

**Weaknesses:**

more comparisons could be helpful.  for example, it could have been interesting to see how bGPFA also compares to infinite gpfa with and without model mismatch similar to the synthetic example presented.

from fig 2b and fig 3b, it does appear that infinite gpfa takes substantially longer to reach convergence.  do the authors expect this difference gets substantially worse with higher latent state dimensionality? it could be helpful to see convergence plots for a dataset that requires higher latent state dimensionalities.

**Questions:**

for fig.2c what expected level of masking for Z was used?  on that topic, it could also be interesting to show how the gap between gpfa and infinite gpfa inference performance varies with the expected value of alpha.  additionally, an inline figure or addition to figure 1 helping to build intuition between numerical values of alpha and the expected number of features could be useful.

is the runtime comparison in Fig.2h for a single EM iteration, or only inference time?

---

> ### Author Response · Authors · 2024-11-20
> **Response to Reviewer swfp**
>
> We thank the reviewer for constructive feedbacks and overall positive
> recognition of our work. Please find responses to the raised questions and
> weaknesses below.
>
> - The main criticism to the submitted manuscript is the absence of
>     baseline comparison, specifically with respect to the Bayesian GPFA model
>     (Jensen et al. 2021). We have implemented and evaluated the Bayesian GPFA
>     model on both the synthetic dataset and the real neural dataset, and have
>     revised the manuscript to include the comparison. The key takeaway for this
>     comparison is summarised as following.
>     - Averaging over multiple random seeds, Bayesian GPFA converges to
>         similar asymptotic free energy value as infinite GPFA, given the
>         synthetic data generated from both generative processes with and without
>         stochastic binary loading mask. However, the Bayesian GPFA training is
>         significantly less stable (indicated by the greater standard deviation
>         across different random seeds, see Figure 2b in the revised manuscript).
>         On the real neural dataset, Bayesian GPFA converges to a lower value as
>         infinite GPFA, and the CCA analysis reveals that the infinite GPFA
>         latents provides more accurate representation of the relevant
>         behavioural variables (Figure 3c).
>     - In addition to the different priors used for automatic model
>         selection (ARD and IBP for Bayesian GPFA and infinite GPFA,
>         respectively), a key difference between the two models lie in the
>         approximate inference. The Bayesian GPFA model leverages circulant
>         approximation to the full covariance matrix in the variational GP
>         distribution, whereas the infinite GPFA model leverages the sparse
>         variational approximation with inducing points. The ability to model the
>         covariance matrix over all input locations enables Bayesian GPFA to
>         capture information on a finer temporal resolution, which explains why
>         the fitted latents directly capture the loaded latents
>         ($\tilde{\mathbf{f}} = \mathbf{Z}\odot\mathbf{f}$; Figure 2e, bottom
>         panel). However, modelling the loaded latents with only the GP latents
>         using the circulant approximation raises concerns. Specifically, direct
>         modelling of the loaded latents lacks interpretability, such that it is
>         difficult to separate variabilities between the latent processes and the
>         loading processes (Figure 2e, bottom panel, and Figure 3c).
>     - The circulant approximation of the full covariance matrix induces
>         greater approximation error as the latent dimension increases (Figure
>         2c, bottom).
>     - The computational complexity for the circulant
>         approximation scales supra linearly with the number of inputs, $T$,
>         ($\mathcal{O}(T\log T)$). The resulting model is prohibitively expensive
>         for practical application (Figure 2h).
> - We appreciate the reviewer's sharp observation that the infinite GPFA
>     model requires more training epochs to reach convergence. We have run
>     additional ablations studies and show that the difference in the amount of
>     training to reach convergence between standard and infinite svGPFA does not
>     increase with the number of latent dimensions in the generative process
>     (Figure S3 a-d in the revised manuscript). In contrast, the performance gap
>     (in terms of asymptotic free energy objective) between the trained models
>     under the two conditions with and without encoding variability grows
>     significantly for the standard svGPFA model as the latent dimension
>     increases, whereas the same gap for the infinite svGPFA model is minimally
>     affected.
> - The expected level of masking, or the expected level of sparsity in
>     the binary masking matrix, $Z$, is $0.6$ in Figure 2 of the main paper. We
>     define $\tilde{\alpha} = \langle\frac{\sum_{n, d}z_{nd}}{ND}\rangle$ for
>     quantifying the expected level of sparsity. We quantify the difference (in
>     percentage) of asymptotic free energy objective for the standard and
>     infinite svGPFA models for varying levels of $\tilde{\alpha}$. From Figure
>     S3f in the revised manuscript, we observe that the gap increases as the
>     degree of sparsity increases (i.e., $\tilde{\alpha}$ decreases).
> - The runtime comparison in Figure 2h in the revised manuscript now
>     shows the average wallclock time for a single EM iteration during training
>     for the different models (the previous version shows the average wallclock
>     time for $2000$ EM iterations during training). Note that the revised Figure
>     2h now includes the comparison with bGPFA.
>
> We again thank the reviewer for their instructive comments and feedbacks, and we
> hope our responses and revised manuscripts have addressed raised concerns. We
> are happy to engage in any extended discussion if there is any remaining
> question.

---

### Official Review · Reviewer_52sh · 2024-11-03

**Soundness:** 3
**Presentation:** 4
**Contribution:** 3
**Rating:** 8
**Confidence:** 3

**Summary:**

The authors propose a novel model, an extension to GPFA, that incorporates stochastic activation of latent factors in the loading process via IBP prior. This results in dynamically switching expression of latent factors for each neuron across different timepoints, hence incorporating the dynamic shifts in internal states of the animal. They apply their model, infinite GPFA, to two datasets, one synthetic and one real world  neuroscience dataset (hippocampal place cells during spatial navigation).

**Strengths:**

- **Novelty**: The proposed model, infinite GPFA, has a robust mechanism that allows for  estimation of both the number of latent factors and their time-varying activations without requiring manual tuning. In addition, the sparsity allows for learning of more interpretable latent factors, which is helpful for interpreting neural computations.
- This framework opens up new avenues in neuroscience for exploratory investigations of experimental data.
- Presentation is clear.

**Weaknesses:**

- More comparison to other methods could have strengthened the utility and performance of infinite GPFA, specifically, using some of the previously established methods like GPFA with ARD prior. Although GPFA with ARD prior is not designed to capture latent factors across time, it would be useful to show it quantitatively.

Minor points
- l060 ‘An as example,’ → ‘As an example,’
- Figure2.a the axis labels are illegible .
- In general figure 2 gets rendered very slowly, I am not sure the exact cause but it might be worth investigating because if it’s simple like rasterization or high resolution graphics, it can be easy to fix.

**Questions:**

- How does the infinite GPFA handle cases where it identifies overlapping/slightly differing  latent factors ?

---

> ### Author Response · Authors · 2024-11-20
> **Response to Reviewer 52sh**
>
> We thank the reviewer for constructive feedbacks and overall positive
> recognition of our work. Please find responses to the raised questions and
> weaknesses below.
>
> - The main criticism to the submitted manuscript is the absence of
>     baseline comparison, specifically with respect to the Bayesian GPFA model
>     (Jensen et al. 2021). We have implemented and evaluated the Bayesian GPFA
>     model on both the synthetic dataset and the real neural dataset, and have
>     revised the manuscript to include the comparison. The key takeaway for this
>     comparison is summarised as following.
>     - Averaging over multiple random seeds, Bayesian GPFA converges to
>         similar asymptotic free energy value as infinite GPFA, given the
>         synthetic data generated from both generative processes with and without
>         stochastic binary loading mask. However, the Bayesian GPFA training is
>         significantly less stable (indicated by the greater standard deviation
>         across different random seeds, see Figure 2b in the revised manuscript).
>         On the real neural dataset, Bayesian GPFA converges to a lower value as
>         infinite GPFA, and the CCA analysis reveals that the infinite GPFA
>         latents provides more accurate representation of the relevant
>         behavioural variables (Figure 3c).
>     - In addition to the different priors used for automatic model
>         selection (ARD and IBP for Bayesian GPFA and infinite GPFA,
>         respectively), a key difference between the two models lie in the
>         approximate inference. The Bayesian GPFA model leverages circulant
>         approximation to the full covariance matrix in the variational GP
>         distribution, whereas the infinite GPFA model leverages the sparse
>         variational approximation with inducing points. The ability to model the
>         covariance matrix over all input locations enables Bayesian GPFA to
>         capture information on a finer temporal resolution, which explains why
>         the fitted latents directly capture the loaded latents
>         ($\tilde{\mathbf{f}} = \mathbf{Z}\odot\mathbf{f}$; Figure 2e, bottom
>         panel). However, modelling the loaded latents with only the GP latents
>         using the circulant approximation raises concerns. Specifically, direct
>         modelling of the loaded latents lacks interpretability, such that it is
>         difficult to separate variabilities between the latent processes and the
>         loading processes (Figure 2e, bottom panel, and Figure 3c).
>     - The circulant approximation of the full covariance matrix induces
>         greater approximation error as the latent dimension increases (Figure
>         2c, bottom).
>     - The computational complexity for the circulant
>         approximation scales supra linearly with the number of inputs, $T$,
>         ($\mathcal{O}(T\log T)$). The resulting model is prohibitively expensive
>         for practical application (Figure 2h).
> - We thank the reviewer for pointing out the typos and previously
>     illegible axis labels in Figure 2a. We have now addressed them in the
>     revised manuscript.
> - We thank the reviewer for noting the figure rendering issue with
>     Figure 2. This is due to high dpi associated with the scatter plots in
>     Figure 2d. We have now replaced the figure with a copy with lower dpi, which
>     should now improve the rendering speed.
> - Regarding the question on how does the infinite GPFA handles overlapping/slightly differing latent factors, we emphasise that in our ablation studies on the synthetic dataset (see Figure S3 in the revised manuscript), some of the additional latent processes exhibit purely phase-shift relative to earlier latents (e.g., sin(3x) and cos(3x)). Under such conditions, the infinite GPFA still infers the ground-truth latents with high accuracy.
>
> We again thank the reviewer for their instructive comments and feedbacks, and we
> hope our responses and revised manuscripts have addressed raised concerns. We
> are happy to engage in any extended discussion if there is any remaining
> question.

---

### Official Review · Reviewer_LfyK · 2024-11-03

**Soundness:** 4
**Presentation:** 3
**Contribution:** 4
**Rating:** 8
**Confidence:** 4

**Summary:**

The authors present an extension to GPFA, a widely used latent variable model in neuroscience, that uses an Indian Buffet process as a nonparametric extension to automatically select latent dimensions at each time point. This avoids the need for a priori latent dimensionality choice in GPFA, a well-known limitation to the method, and allows for a sparse selection of latent activations at each time point, which can identify transitions in the latent representation, enhancing the models usefulness in the identification of behavioral states. The authors show strong validation on synthetic datasets as well as real spiking data. The theory is clear and model development and implementation is clear and sound.

**Strengths:**

The switching GPFA and switching infinite GPFA models effectively tackle a significant limitation commonly encountered in many latent variable models in neuroscience, particularly within GPFA: the a priori selection of latent dimensionality. Additionally, these models enhance the approach by allowing for unequal contributions of latent variables at different time points, addressing another critical shortcoming of traditional GPFA. This advancement represents a noteworthy contribution to latent variable modeling in neuroscience. The authors also incorporate inducing points for improved scalability, a practical and well-established extension from the existing GP literature.

**Weaknesses:**

The weakest part of the manuscript is the lack of evaluations to any competing approach. The authors appear only compare to variants of their own model. In particular, because the authors emphasize the advantage of not needing to pre-select latent dimensionality, some evaluation against the ARD approach in Jensen et al would be appreciated. The authors claim the ARD is inferior due to requiring marginalizing over all of the data to determine latent dimensionality, and this is sound reasoning, however, I am curious as to how exactly different the models fits and latent posteriors would be. It might be possible, for example, for the ARD GPFA model to learn an appropriate number of latent dimensions and have periods of time where different groups of latents are minimally or highly variable. I think it would help a reader get a sense of how svGPFA compares Bayesian GPFA, as the latter is a model that was motivated in a very similar way.

Note also that the manuscript "Uncovering motifs of concurrent signaling across multiple neuronal populations", Gokcen et al. also uses an ARD prior in a similar GPFA-style model - might be worth citing

One small point -- Figure 2 is difficult to render in the browser and this specific page lags. I suspect the figure size is too large, maybe due to panel d. Downsampling this figure before adding it to the latex might help.

**Questions:**

None

---

> ### Author Response · Authors · 2024-11-20
> **Response to Reviewer LfyK**
>
> We thank the reviewer for constructive feedbacks and overall positive
> recognition of our work. Please find responses to the raised questions and
> weaknesses below.
>
> - The main criticism to the submitted manuscript is the absence of
>     baseline comparison, specifically with respect to the Bayesian GPFA model
>     (Jensen et al. 2021). We have implemented and evaluated the Bayesian GPFA
>     model on both the synthetic dataset and the real neural dataset, and have
>     revised the manuscript to include the comparison. The key takeaway for this
>     comparison is summarised as following.
>     - Averaging over multiple random seeds, Bayesian GPFA converges to
>         similar asymptotic free energy value as infinite GPFA, given the
>         synthetic data generated from both generative processes with and without
>         stochastic binary loading mask. However, the Bayesian GPFA training is
>         significantly less stable (indicated by the greater standard deviation
>         across different random seeds, see Figure 2b in the revised manuscript).
>         On the real neural dataset, Bayesian GPFA converges to a lower value as
>         infinite GPFA, and the CCA analysis reveals that the infinite GPFA
>         latents provides more accurate representation of the relevant
>         behavioural variables (Figure 3c).
>     - In addition to the different priors used for automatic model
>         selection (ARD and IBP for Bayesian GPFA and infinite GPFA,
>         respectively), a key difference between the two models lie in the
>         approximate inference. The Bayesian GPFA model leverages circulant
>         approximation to the full covariance matrix in the variational GP
>         distribution, whereas the infinite GPFA model leverages the sparse
>         variational approximation with inducing points. The ability to model the
>         covariance matrix over all input locations enables Bayesian GPFA to
>         capture information on a finer temporal resolution, which explains why
>         the fitted latents directly capture the loaded latents
>         ($\tilde{\mathbf{f}} = \mathbf{Z}\odot\mathbf{f}$; Figure 2e, bottom
>         panel). However, modelling the loaded latents with only the GP latents
>         using the circulant approximation raises concerns. Specifically, direct
>         modelling of the loaded latents lacks interpretability, such that it is
>         difficult to separate variabilities between the latent processes and the
>         loading processes (Figure 2e, bottom panel, and Figure 3c).
>     - The circulant approximation of the full covariance matrix induces
>         greater approximation error as the latent dimension increases (Figure
>         2c, bottom).
>     - The computational complexity for the circulant
>         approximation scales supra linearly with the number of inputs, $T$,
>         ($\mathcal{O}(T\log T)$). The resulting model is prohibitively expensive
>         for practical application (Figure 2h).
> - We thank the reviewer for noting our current omission of citing the
>     relevant Gokcen et al. study, we have now cited the paper in the revised
>     manuscript with accompanying discussion.
> - We thank the reviewer for noting the figure rendering issue with
>     Figure 2. This is due to high dpi associated with the scatter plots in
>     Figure 2d. We have now replaced the figure with a copy with lower dpi, which
>     should now improve the rendering speed.
>
> We again thank the reviewer for their instructive comments and feedbacks, and we
> hope our responses and revised manuscripts have addressed raised concerns. We
> are happy to engage in any extended discussion if there is any remaining
> question. If all raised concerns are resolved satisfactorily, we sincerely hope the reviewer could raise their score accordingly.

---

> > ### Author Response · Authors · 2024-11-25
> > **Thank you for raising the score**
> >
> > We glad to see that our responses have successfully addressed the reviewer's comments and questions, and we wish to thank the reviewer for raising their score. Please do let us know if there is any question remaining.

---

### Official Review · Reviewer_iwdZ · 2024-11-04

**Soundness:** 3
**Presentation:** 3
**Contribution:** 2
**Rating:** 6
**Confidence:** 3

**Summary:**

This paper proposes the infinite GPFA model, which is Bayesian non-parametric extension of the classic GPFA by combining GPFA with an Indian Buffet Process Prior. This model can potentially infer infinite set of latent factors from data. A variational EM algorithm is proposed to perform the inference. The authors demonstrate the effectiveness of this model through analysis on simulated and real datasets.

**Strengths:**

- The paper is clearly written. The model formulation and related works are clearly introduced.
- The authors have done extensive experiments on real neural data and synthetic data, and results seem good.

**Weaknesses:**

- The idea of combining GPFA with IBP prior is not revolutionary.
- I listed some questions in the section below.

**Questions:**

- unclear sentence line 366: "Moreover, in an SLDS, only the latent dynamics changes following context switching, hence requiring a non-negligible number of timesteps (depending on the spectral radius of transition operator) for the reflection of context changes in the observation space. in contrast, the compositional nature of factor loading process in the infinite GPFA model allows immediate differential expression of latent processes into neural activities."

Can you clarify this a bit? infinite GPFA model seems to also have the factor loading process in latent space, why it allows immediate expression into neural activities than SLDS?

- How is the number of features D selected for svGPFA in the experiments section for synthetic data and real data?

- What's the future research direction for this paper?

---

> ### Author Response · Authors · 2024-11-20
> **Response to Reviewer iwdZ**
>
> We thank the reviewer for constructive feedbacks and overall positive
> recognition of our work. Please find responses to the raised questions and
> weaknesses below.
>
> - Regarding the reviewer's concern that the combination of GPFA with IBP
>     prior is not revolutionary, we wish to emphasise that the infinite GPFA
>     model is, to the best of our knowledge, the first model that explicitly
>     incorporates the IBP prior in latent variable models with continuous-time
>     latents. Hence, from the methodological perspective, we believe the idea
>     behind the infinite GPFA model contains sufficient novelty. We understand
>     that it is possible there are existing models that exhibit the above model
>     features, and we respectfully ask the reviewer to inform us of such model,
>     and we are happy to cite them in the revised manuscript. From the neural
>     data analysis perspective, the infinite GPFA model provides a completely new
>     way of interpreting population neuronal firing: the expression/encoding of
>     latent (behavioural) information in both single-neuron and population
>     activities is potentially not constant over time. The novel interpretation
>     of neural coding enables us to better understand transient changes in neural
>     activities (see, e.g., discussions regarding the motivations from the
>     neuroscience perspective in l.72-l.80 and the Discussion section in the
>     revised manuscript).
> - The infinite GPFA model models the firing rate of each neuron (in
>     log-space) as the weighted product of the binary loading ($Z$) and the
>     latent processes ($f$). Binary expression of latent processes in the neural
>     activities might happen at a might higher temporal scale than the temporal
>     variations in the latent processes themselves (see, e.g., the discussion on
>     exemplary datasets that exhibit such property from Kelemen and Fenton, 2010,
>     and Jezek et al. 2011, in Section 1 of the revised manuscript). Hence,
>     changes in transition alone (as in SLDS models) is not able to promptly
>     reflect such contextual changes in the neural responses within the
>     biological plausible timescale. We have now updated our discussion regarding
>     comparison with SLDS models in the Related Works section in the revised
>     manuscript to improve its clarity. We hope these additional statements have
>     clearly conveyed our points, but we are happy to engage in further
>     discussions if there is any point remains unclear.
> - The number of feature, $D$, is identical for both standard and
>     infinite svGPFA (and our new baseline comparison, the bGPFA model) for
>     experiments with both synthetic and real datasets. Specifically, we set $D =
>     10$ for both the synthetic dataset and the real neural data. These
>     hyperparameters are provided in Supplemental S3.1 in the revised manuscript.
> - There are multiple future directions for the infinite GPFA model. From
>     the methodological perspective, it is possible to extend the model to
>     include non-linear generative and recognition networks to improve the
>     modelling flexibility of the latent variable model (potentially under the
>     variational autoencoder framework, see, e.g., Yu, et al. 2022). Moreover,
>     despite introducing temporally varying loading process, the loading matrix,
>     $C$, is still assumed to be stationary. Hence, another possible extension to
>     the infinite GPFA model would be incorporating priors over the loading
>     weight matrix $C$ with non-trivial temporal dependency (e.g., GP). From a
>     neuroscience perspective, due to the linear assumption of the proposed
>     model, it might be more suitable for modelling neural recordings from motor
>     cortex, which have been shown to exhibit strongly linear tuning properties
>     (see, e.g., Paninski et al. 2004 and Yu et al. 2008), and we leave this for
>     future studies.
>
> We again thank the reviewer for their instructive comments and feedbacks, and we
> hope our responses and revised manuscripts have addressed raised concerns. We
> are happy to engage in any extended discussion if there is any remaining
> question. If all raised concerns are resolved satisfactorily, we sincerely hope
> the reviewer could raise their score accordingly.
>
> References:
>
> Yu, C., Soulat, H., Burgess, N. and Sahani, M., 2022. Structured recognition for generative models with explaining away. Advances in Neural Information Processing Systems, 35, pp.40-53.
>
> Paninski, L., Fellows, M.R., Hatsopoulos, N.G. and Donoghue, J.P., 2004. Spatiotemporal tuning of motor cortical neurons for hand position and velocity. Journal of neurophysiology, 91(1), pp.515-532.
>
> Yu, B.M., Cunningham, J.P., Santhanam, G., Ryu, S., Shenoy, K.V. and Sahani, M., 2008. Gaussian-process factor analysis for low-dimensional single-trial analysis of neural population activity. Advances in neural information processing systems, 21.

---

> > ### Author Response · Authors · 2024-11-25
> > **Follow-up on our responses**
> >
> > We wish to kindly remind the reviewer that the end of the rebuttal period is fast approaching. We are happy to address any outstanding comment raised by the reviewer. In the meantime, we hope our responses above has adequately addressed previously raised questions/comments, and the reviewer could update their score accordingly if there is no additional question/comment remaining.

---

> > > ### Comment · Reviewer_iwdZ · 2024-12-03
> > >
> > > I thank the authors for their responses, and I'll maintain my score.

---

### Author Response · Authors · 2024-11-20
**General Response to All Reviewers**

We thank all reviewers for their constructive feedbacks and overall positive
recognition of our paper and the infinite GPFA model. Here we provide some
general responses to some questions raised by multiple reviewers.

- **Lack of baseline comparison.** We note the main criticism to
    the submitted subscript is the lack of baseline comparison, specificially
    with respect to the Bayesian GPFA model (Jensen et al. 2021). We have
    implemented and evaluated the Bayesian GPFA model on both the synthetic
    dataset and the real neural dataset, and have revised the manuscript to
    include the comparison. The key takeaway for this comparison is summarised
    as following.
    - Averaging over multiple random seeds, Bayesian GPFA converges to
        similar asymptotic free energy value as infinite GPFA, given the
        synthetic data generated from both generative processes with and without
        stochastic binary loading mask. However, the Bayesian GPFA training is
        significantly less stable (indicated by the greater standard deviation
        across different random seeds, see Figure 2b in the revised manuscript).
        On the real neural dataset, Bayesian GPFA converges to a lower value as
        infinite GPFA, and the CCA analysis reveals that the infinite GPFA
        latents provides more accurate representation of the relevant
        behavioural variables (Figure 3c).
    - In addition to the different priors used for automatic model
        selection (ARD and IBP for Bayesian GPFA and infinite GPFA,
        respectively), a key difference between the two models lie in the
        approximate inference. The Bayesian GPFA model leverages circulant
        approximation to the full covariance matrix in the variational GP
        distribution, whereas the infinite GPFA model leverages the sparse
        variational approximation with inducing points. The ability to model the
        covariance matrix over all input locations enables Bayesian GPFA to
        capture information on a finer temporal resolution, which explains why
        the fitted latents directly capture the loaded latents
        ($\tilde{\mathbf{f}} = \mathbf{Z}\odot\mathbf{f}$; Figure 2e, bottom
        panel). However, modelling the loaded latents with only the GP latents
        using the circulant approximation raises concerns. Specifically, direct
        modelling of the loaded latents lacks interpretability, such that it is
        difficult to separate variabilities between the latent processes and the
        loading processes (Figure 2e, bottom panel, and Figure 3c).
    - The circulant approximation of the full covariance matrix induces
        greater approximation error as the latent dimension increases (Figure
        2c, bottom).
    - The computational complexity for the circulant
        approximation scales supra linearly with the number of inputs, $T$,
        ($\mathcal{O}(T\log T)$). The resulting model is prohibitively expensive
        for practical application (Figure 2h).
- **Rendering Issue with Figure 2d.** As pointed out by multiple
    reviewers, the rendering of Figure 2 is prohibitively slow. This is due to
    high dpi associated with the scatter plots in Figure 2d. We have now
    replaced the figure with a copy with lower dpi, which should now improve the
    rendering speed.

We again thank all reviewers for their instructive comments and feedbacks, and
we hope our responses and revised manuscripts have addressed all raised
concerns. We are happy to engage in any extended discussion if there is any
remaining question.

---

### Meta-Review · Area_Chair_6QTR · 2024-12-17

**Metareview:**

This paper proposes the infinite GPFA model, a Bayesian nonparametric extension of the classical GPFA that combines GPFA with an Indian Buffet Process (IBP) prior. The model allows for the inference of a potentially infinite set of latent factors without requiring an a priori choice of latent dimensionality and enables dynamic activation of latent factors over time. A variational EM algorithm is introduced for inference, and the model is validated through synthetic and real neural datasets, demonstrating its ability to reveal behaviorally relevant latent structures.

The infinite GPFA effectively addresses key limitations of classical GPFA, such as fixed latent dimensionality and the inability to model time-varying latent activations. The use of IBP introduces sparsity, enhancing interpretability of the latent factors and enabling identification of dynamic shifts in neural states. The authors provide clear model formulation, practical variational inference, and thorough validation on both synthetic and real datasets, demonstrating the method's utility in uncovering latent neural dynamics.

The main weakness is the lack of comparisons to alternative methods, such as GPFA with Automatic Relevance Determination (ARD), which could help better contextualize the model's improvements. Additionally, runtime analysis reveals that the infinite GPFA requires longer convergence times, and providing more details on scalability would strengthen the evaluation. While most concerns were addressed in the authors' rebuttal, I appreciate the helpful feedback from the reviewers and the authors' responses. However, I believe many reviewers may not fully recognize that using IBP for sparse feature selection is a standard approach in non-parametric Bayesian methods. Pearce et al. (2017) already applied IBP to factor analysis, and equation 2 in that paper is essentially the same as equation 3 in this paper. The primary novelty here lies in the introduction of a GP prior over the latent variables, which leads to interesting neuroscience results. While this is a valuable contribution to the neuroscience field, I believe the level of novelty and intellectual merit may not meet the standards expected for ICLR. However, given that the reviewers all maintain a high level of enthusiasm, I recommend for acceptance.

**Additional Comments On Reviewer Discussion:**

The primary concern raised by nearly all reviewers was the lack of comparison. The author has addressed this by implementing and evaluating the Bayesian GPFA model on both synthetic and real neural datasets and revising the manuscript to include the comparison. Considering that the review scores are generally high and no other major issues have been identified, I recommend acceptance.

---

### Decision · Program_Chairs · 2025-01-22

Accept (Spotlight)